# A new opportunity for the emerging tellurium semiconductor: making resistive switching devices

Yifei Yang [1,11], Mingkun Xu[1,11], Shujing Jia[2,3,11], Bolun Wang[4], Lujie Xu [5], Xinxin Wang[1], Huan Liu[6], Yuanshuang Liu[6], Yuzheng Guo [7], Lidan Wang[8], Shukai Duan[9], Kai Liu [4], Min Zhu [3], Jing Pei[1✉], Wenrui Duan[5✉], Dameng Liu[6✉] & Huanglong Li [1,10✉]

The development of the resistive switching cross-point array as the next-generation platform for high-density storage, in-memory computing and neuromorphic computing heavily relies on the improvement of the two component devices, volatile selector and nonvolatile memory, which have distinct operating current requirements. The perennial current-volatility dilemma that has been widely faced in various device implementations remains a major bottleneck. Here, we show that the device based on electrochemically active, low-thermal conductivity and low-melting temperature semiconducting tellurium filament can solve this dilemma, being able to function as either selector or memory in respective desired current ranges. Furthermore, we demonstrate one-selector-one-resistor behavior in a tandem of two identical Te-based devices, indicating the potential of Te-based device as a universal array building block. These nonconventional phenomena can be understood from a combination of unique electrical-thermal properties in Te. Preliminary device optimization efforts also indicate large and unique design space for Te-based resistive switching devices.

[1] Department of Precision Instrument, Center for Brain Inspired Computing Research, Tsinghua University, Beijing 100084, China. [2] Frontier Institute of Chip and System, Fudan University, Shanghai 200433, China. [3] State Key Laboratory of Functional Materials for Informatics, Shanghai Institute of Micro-System and Information Technology, Chinese Academy of Sciences, Shanghai 200050, China. [4] State Key Laboratory of New Ceramics and Fine Processing, School of Materials Science and Engineering, Tsinghua University, Beijing 100084, China. [5] School of Instrument Science and Opto Electronics Engineering, Beijing Information Science & Technology University, Beijing 100101, China. [6] State Key Laboratory of Tribology, Tsinghua University, Beijing 100084, China. [7] College of Engineering, Swansea University, SA1 8EN Swansea, UK. [8] School of Electronic and Information Engineering, Southwest University, Chongqing 400715, China. [9] School of Artificial Intelligence, Southwest University, Chongqing 400715, China. [10] Chinese Institute for Brain Research, Beijing 102206, China. [11] These authors contributed equally: Yifei Yang, Mingkun Xu, Shujing Jia. ✉email: peij@tsinghua.edu.cn; duanwr@mail.tsinghua.edu.cn; ldm@tsinghua.edu.cn; li_huanglong@mail.tsinghua.edu.cn

The resistive switching (RS) cross-point arrays are emerging technologies for high-density data storage and non-conventional information processing, such as in-memory computing, neuromorphic computing, and machine learning[1–4]. For large-scale arrays, each cross-point consists of a nonvolatile RS (NV-RS) memory device, and in series, a three-terminal transistor or a two-terminal volatile RS (V-RS) selector device to suppress the undesired sneak-path currents. So far, although transistors enable the most reliable array operations, V-RS selectors are of great promise for maximizing the density of integration[5,6]. There are various types of NV-RS memories based on a broad category of materials and physical mechanisms[7], including phase change memories[8], ferroelectric memories[9], magnetoresistive memories[10], valance change memories[11], electrochemical (EC)-RS memories[12], and so on. As for V-RS selectors, ovonic threshold switches (OTS)[13,14], Mott selectors[15,16], EC-RS selectors[17], and so on have gained considerable research interests. Among these device technologies, EC-RS devices are particularly attractive in terms of the simplicity of the working principles and the diversity of functions, used for NV-RS memories[18–20], V-RS selectors[21], and neuronal emulators[22].

In addition to the different switching behaviors, NV-RS memories and V-RS selectors also have distinct operating requirements: the operating currents of NV-RS memories need to be as low as possible to minimize power consumption, while V-RS selectors should be able to operate under high ON-state currents to ensure successful writing to the memories and provide sufficient read margins[23–25]. Considerable progresses have been made recently to satisfy these requirements that NV-RS memories with operating currents as low as 10 pA[26–29], and chalcogenide-based OTS V-RS selectors with ON-state current densities exceeding 20 MA/cm² [13,14,30] have been demonstrated.

Despite these progresses, a long-standing dilemma, known as the current-volatility dilemma, which has been widely faced in various implementations of NV-RS memories and V-RS selectors, has remained a major bottleneck for further performance improvement of the devices. To be specific, the current-volatility dilemma is a phenomenon that for an RS device the V-RS normally occurs under lower operating current than does the NV-RS. The current under which the V-RS-to-NV-RS transition takes place sets a fundamental limit on the highest and lowest currents under which the device can operate in the V-RS and NV-RS modes, respectively. This prevents the operating current of the device as either selector or memory from going even higher or lower, respectively, as desired.

So far, the difficulty brought about by this dilemma in further improving the device performance can be tactfully bypassed through resorting to new materials and (or) new RS mechanisms that either push the transition currents higher or lower for better selectors or memories, respectively. For example, Zhao et al.[31] introduced graphene with different structure defects to the Ag/SiO₂-based EC-RS devices and realized low and high operating currents for NV-RS memories and V-RS selectors, respectively. However, this was achieved at the expense of fabrication complexity that graphene must be engineered to different forms in order to satisfy the respective operating current requirements for memories and selectors. In other EC-RS devices, either V-RS under high currents[32] or NV-RS under low currents[29] has been demonstrated. However, as just mentioned, the current-volatility dilemma has been bypassed but has never really been solved.

For EC-RS devices, the current-volatility dilemma is seemingly fundamental as it can be understood from the fact that the stabilities of the filaments increase with the strengths of the EC effects and therefore with the operating currents[33]. In addition to the EC effect, electrical current of course induces Joule heat. As a conventional wisdom, however, the Joule heating (JH) effect in the EC-RS device is secondary and synergetic that it assists the lateral diffusion of atoms or ions, resulting in the increase of the thickness of the filament and therefore its stability. In light of this, a clue to the solution to this dilemma is using effects that can counteract the EC effect in filament growth, reversing the current dependence of filament stability. As just mentioned, the JH effect could be such a candidate effect. To this end, filaments composed of materials with low melting temperatures are useful because they might be ruptured when JH effects become sufficiently strong. In addition, filamentary materials with low thermal conductivities are more likely to confine heat and facilitate heat accumulation. In addition, of course, new materials should also be electrochemically active to enable filament growth.

Currently, main reported filamentary materials for EC-RS devices are all metals with relatively high thermal conductivities and high melting temperatures[12]. Recently, tellurium (Te), as an emerging semiconductor material for the next-generation transistors[34–37], has been found to be electrochemically active[38–43]. Associated with its semiconductivity, Te also has low thermal conductivity ($1.6 \, W \, m^{-1} \, K^{-1}$)[44] compared to those of the metallic filamentary materials (e.g., Cu: $401 \, W \, m^{-1} \, K^{-1}$; Ag: $429 \, W \, m^{-1} \, K^{-1}$)[45]. This has been one of the main reasons behind its attractive thermoelectric performance[44]. In addition, Te has the second lowest melting point (452 °C)[46] among all elemental semiconductors. These combined properties as desired endow Te with the aptitude to be the appropriate filamentary material.

In this work, we demonstrate RS devices based on semiconducting Te filaments. Unlike conventional EC-RS devices based on metallic filaments, our devices show opposite behavior that the NV-RS occurs under lower operating current than does the V-RS. Moreover, we show that the operating current limit can be pushed (for $2 \times 2$-μm² devices) downward to several μA in the NV-RS mode and upward to several mA in the V-RS mode, truly solving the current-volatility dilemma. We also find that in a tandem of two identical Te-based RS devices the typical one-selector–one-resistor (1S1R) behavior can be reproduced. This demonstrates the potential of Te-based device as a universal building block for the RS cross-point array. These phenomena can be understood from the transition of the EC-JH relationship from synergetic to adversarial as the current increases, which also reverses the current dependence of filament stability. Under pulse train measurements, by varying the degree of rivalry between the EC and JH effects, unusual long-term plasticity (LTP) to short-term plasticity (STP) transition under high pulse frequency is observed, which is in contrast to the commonly observed STP-to-LTP transition in EC-RS devices. This device character could be utilized in the spatial-temporal signal filter layers in spiking neural networks (SNNs) for high-performance event-based visual recognition tasks, as demonstrated in our noise filtering simulations. In an attempt to further optimize our devices, we realize that the design space of Te-based RS devices is potentially large and unique that even the protective electrodes with different work functions and the dielectrics with different thermal conductivities can dramatically affect the device behavior. A combination of unique electrical-thermal properties makes Te an attractive and promising enabler for future RS devices.

## Results and discussion

**Resistive switching behaviors of Te-based devices.** The transmission electron microscopy (TEM) image of the cross-section of the Te/Sb₂Te₃/Te (TST) RS device is shown in Fig. 1a. The results of the elemental mapping by energy-dispersive X-ray spectroscopy (EDS) line scanning along the yellow line as denoted in Fig. 1a are shown in Fig. 1b and Supplementary Fig. S1. It is seen that the deposited Sb₂Te₃ film is close to its stoichiometry, with 39.4% of Sb and 60.6% of Te.

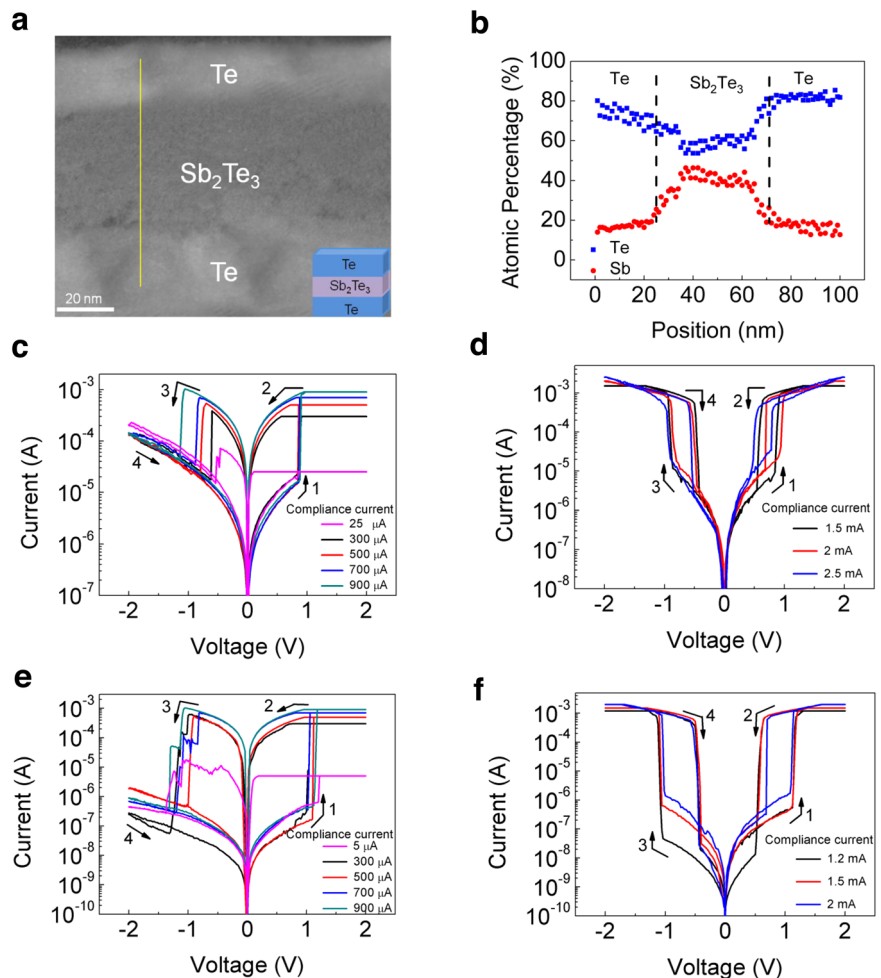

**Fig. 1 Device structure and RS characteristics. (a)** TEM image of the cross-section of the TST device. (**b**) Distribution of elements analyzed by EDS line scan along the yellow line denoted in (**a**). I–V curves of the Pt-protected TST device (**c**) in its NV-RS mode under various CCs (**d**) and in its V-RS mode under various CCs. I–V curves of the Gd protected TST device (**e**) in its NV-RS mode under various CCs (**f**) and in its V-RS mode under various CCs.

The $\log|I|$–$V$ curves for the RS processes in a $2 \times 2$-$\mu m^2$ TST device under five typical compliance currents (CCs) below 1 mA are shown in Fig. 1c. The measured current under the positive voltage sweep (from 0 V) is sharply increased to the CC limit at ~1 V and the device reaches its low-resistance state (LRS). This is defined as the SET process (arrow 1). Then, the device maintains its LRS under the backward voltage sweep even after the voltage has ceased (arrow 2), resulting in counter-clockwise hysteretic $\log|I|$–$V$ loop. As the backward voltage sweep continues, the voltage polarity is reversed. When the negative voltage reaches ~ −1 V, the current rapidly drops to a low value, switching the device back to its high-resistance state (HRS). This is defined as the RESET process (arrow 3). It should be pointed out that the retention of the LRS is a prerequisite for the occurrence of the RESET switching. The HRS of the device is then retained under the voltage sweep back to zero again (arrow 4), resulting in another counter-clockwise hysteretic $\log|I|$–$V$ loop. Such $\log|I|$–$V$ curves reflect the typical characteristics of the NV-RS. The long retention of the LRS in the absence of voltage is further confirmed by delayed RESET test and DC stress test, as shown in Supplementary Fig. S2.

Interestingly, when the applied CC is increased to ~1.5 mA, a transition from NV-RS to V-RS occurs. As illustrated in Fig. 1d, the device is first switched to the LRS at ~1 V under forward voltage sweep (arrow 1), similar to the SET process in NV-RS but commonly called the threshold switching (TS) process in V-RS.

Unlike NV-RS, the LRS obtained in V-RS can only be retained when a sufficiently large hold voltage is applied. Under the backward voltage sweep, a dramatic drop of the current occurs at ~0.6 V (arrow 2), indicating the transition from LRS to HRS. As the backward voltage sweep continues, the voltage polarity is reversed. When the voltage becomes sufficiently negative, the current increases sharply (arrow 3), switching the device to its LRS again. This looks like a mirror process of the TS process under the positive voltage polarity. Similarly, this LRS is not stable either and will return to the HRS as the voltage is swept back to a less negative value (arrow 4). The resulting hysteretic $\log|I|$–$V$ loop is clockwise and becomes a mirror loop of the counter-clockwise one under the positive voltage polarity.

Devices with V-RS and NV-RS behaviors could be considered for selector and memory applications, respectively, in the 1S1R cross-point array. As previously introduced, low-operating current memory and high-ON current selector are desired. In this regard, our device is potentially superior in both of these two aspects. As shown in Fig. 1c, we demonstrate that the operating current limit of our device in the NV-RS mode can be pushed downward to 25 µA. This value is lower than that (30 µA) of the state-of-the-art Ag filament-based EC-RS memory device ($5 \times 5$ µm$^2$) with substantial dielectric optimization[47]. This can be understood from the relatively low conductivity of the semiconducting Te filament compared to metallic filaments. On the other hand, Fig. 1d shows that the ON current of our device

in the V-RS mode can reach 2.5 mA. Compared to the state-of-the-art Ag-based EC-RS selector device (100 μA)[21] with larger electrode size ($5 \times 5$ μm$^2$), the drive current of our device is a few tens of times larger. As will be lately introduced, the conduction mechanism of our device in its LRS is filament conduction, therefore the ON current is electrode size-independent. If scaled down from the present μm scale to some tens of nm scale, our device could in principle delivery a few tens of MA/cm$^2$ current density, rivalling the state-of-the-art nanoscale OTS selectors[13,14,30].

As can be seen from Fig. 1c, our proof-of-concept device has relatively high OFF current which sets a fundamental limit on the lowest SET switching current that can be achieved and prevents the on/off ratio from going high. This is due to the relatively high conductivity of the Sb$_2$Te$_3$ dielectric which is known to be a low-bandgap p-type semiconductor. As will be lately introduced, Sb$_2$Te$_3$ has been chosen as the dielectric in consideration of its thermal conductivity in between those of the other two control dielectrics, i.e., Bi$_2$Te$_3$ and TiTe$_2$, to validate our assumption of the adversarial EC-JH in the design, as well as its desired composition, for instance no oxygen, to avoid the possibilities of forming filaments other than Te.

To overcome this limitation, we demonstrate a unique method to reduce the OFF-state conductivity of our device, i.e., protective electrode (PE) engineering. Before we proceed, we want to point out that our TST device is sandwiched between and protected by a pair of Pt electrodes (Pt/Te/Sb$_2$Te$_3$/Te/Pt, or PTSTP). We find that the replacement of the Pt PEs with Gd PEs (GTSTG) can increase the on/off ratio by nearly 100 times, as seen from Fig. 1e. This is related to the semiconducting property of Te. To be specific, it is known that Te is generally p-type and therefore the use of lower-work function Gd PE (2.9 eV) compared to Pt (5.9 eV) increases the Schottky barrier height at the PE/Te interface and consequently the contact resistance. Consistent with our expectation, this also pushes the SET switching current limit further down to 5 μA. This value is comparable to that (1 μA) of the recently reported rare-earth Ru filament-based RS device ($5 \times 5$ μm$^2$)[48]. In its V-RS mode, GTSTG device also has lower OFF-state conductivity compared to that of PTSTP device, as shown in Fig. 1f, due to the high contact resistance at the Gd/Te interface.

Further decrease of the OFF-state conductivity can be achieved by down-scaling the device to nm scale at which the SET switching current could become even lower[29]. T-shape TiN/Te/Sb$_2$Te$_3$/Te/TiN (T′TSTT′) device of the diameter of 60 nm is fabricated by e-beam lithography. Ultralow SET switching current of 50 pA can be achieved, as shown in Supplementary Fig. S3. It is anticipated that by replacing Sb$_2$Te$_3$ with other wide-bandgap insulator, such as SiO$_2$[29] and Ta$_2$O$_5$[48], the OFF-state conductivity and consequently the SET switching current can be further reduced.

Endurance tests have also been performed for both PTSTP and GTSTG devices in their respective NV-RS and V-RS operating modes, as shown in Supplementary Figs. S4 and S5. Continuous quasi-DC sweeps and pulse train measurements show certain degree of on/off ratio degradation. Similar phenomena have also been frequently observed in conventional Ag and Cu-based ECM cells and have generally been attributed to filament overgrowth[49]. In this respect, it is interesting to find that GTSTG device has better endurance performance than PTSTP device. It is likely that the better performance of Gd protected device originates from the smaller electronegativity of Gd (1.21) compared to Pt (2.28) and the consequent stronger binding to Te. This may suppress the injection of excessive Te$^{2-}$ into the dielectric and therefore mitigate filament overgrowth.

In addition to this nonconventional PE engineering method, there are at least two other possible solutions to optimize the endurance performance of the Te-based RS device, i.e., scaling down the device and optimizing the dielectric layer. Currently, our device is $2 \times 2$ μm$^2$. Scaling it further down may limit Te supply and confine Te$^{2-}$ injection into the dielectric layer which may improve its endurance[50]. In addition to spatial confinement of Te$^{2-}$ injection, optimizing the dielectric layer by using less Te-dissolvable dielectric or inserting appropriate Te$^{2-}$ ion buffer layer[49,51–53] is also considered as a viable solution. Here, we investigate the effects of scaling. Supplementary Fig. S6 shows the endurance performance of a 150-nm T′TSTT′ device under pulse train stimuli. As expected[50], the endurance of the down-scaled device enhances compared to the μm-scale device, not only because of the larger dynamic range that is more resistance degradation tolerable but also because of the more spatially confined Te$^{2-}$ injection into the dielectric.

**Resistive switching mechanism of Te-based device.** To understand the mechanism of RS, we first note that Sb$_2$Te$_3$ is a well-known chalcogenide phase change material (PCM) whose solid phase transition between the amorphous state and the crystalline state results in NV-RS. However, our device is not likely to have such phase change-type switching because of the much higher required current, namely, 0.4 A for the $2 \times 2$-μm$^2$ device (extrapolated from the reported RESET current density value for a typical phase change device with much smaller size)[38], than the maximum CC under our test. Actually, PCMs have often been utilized as the dielectric layers in EC-RS devices but most of these devices used electrochemically active Ag or Cu electrodes[54–57]. Recently, the use of Te electrodes has also been found to enable RS[38–43], in which the formation of local Te conducting filaments led to HRS-to-LRS transition while the rupture of filaments led to the reverse transition. In order to confirm whether the RS phenomena in our device are based on such mechanism, we measure the electrode area-dependent resistances of our device in both the NV-RS and V-RS modes. As shown in Fig. 2a, the off-state resistance decreases with increasing area, whereas the on-state resistance is almost independent of the electrode area. These strongly indicate that the RS is originated from the formation and rupture of the localized conducting filaments.

Temperature-dependent conductivity measurements are also conducted on two devices in their respective SET states, i.e., TST device and Ag/Sb$_2$Te$_3$/Ag (ASA) control device. As shown in Supplementary Fig. S7, the conductivity of the TST device in its SET state increases with temperature, indicating the semiconducting property of the local filaments. In contrast, the ASA device shows negative correlation between its SET-state conductivity and temperature, indicating the formation of metallic Ag filaments as well known for such conventional types of devices.

TEM observations of two TST devices after NV-SET switching and V-TS, respectively, are also carried out. The position at which the filamentary switching has taken place is identified after extensive examinations of the 2-μm-wide focused ion beam sample. As compared to the uniform dielectric in the pristine device (Supplementary Fig. S8), a distinct cone-shaped region of the width of 5 nm, connecting the top and bottom electrodes, appears in the dielectric of the NV-SET switched device, as seen in Fig. 2b. EDS elemental mapping reveals that this region is Te-rich (~82.3%). For the V-TS sample, instead of continuously connected filament, dispersive Te-rich nanoclusters in the dielectric are observed, as seen in Fig. 2c. These nanoclusters are expected to be the remains of a ruptured filament.

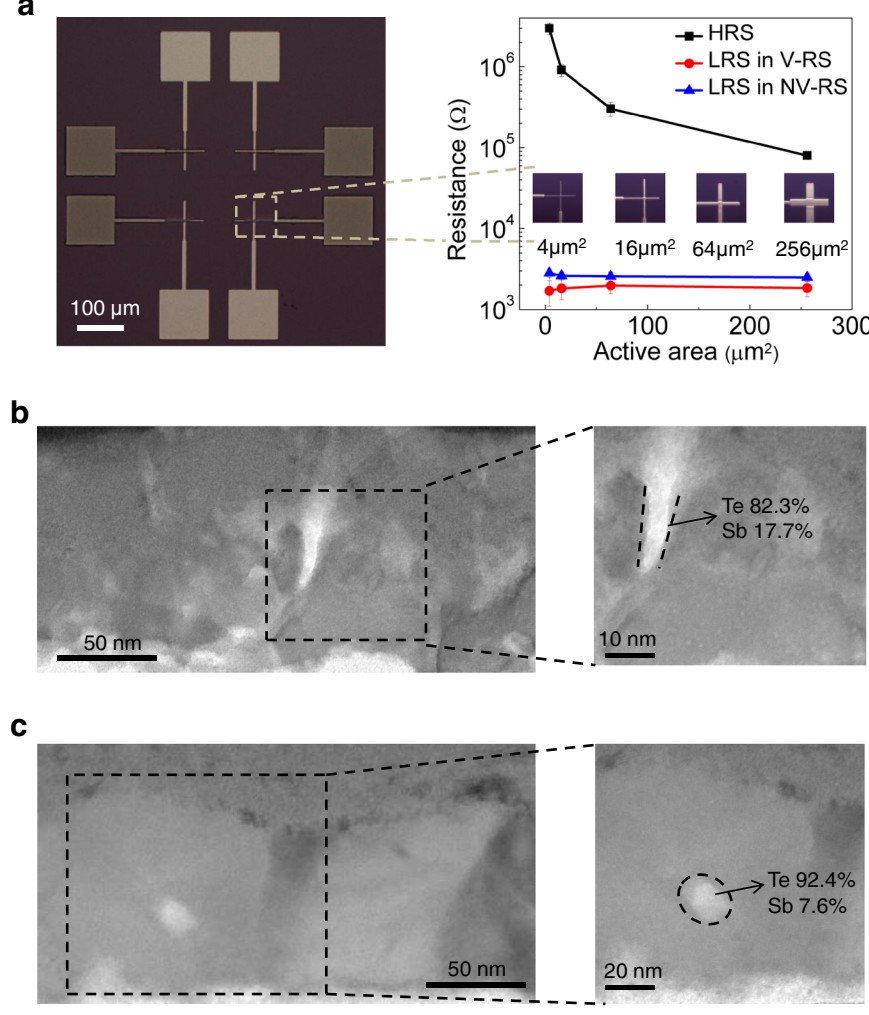

**Fig. 2 Electrode size-independent filamentary conduction and filament observation.** (**a**) Dependence of the resistance on the electrode area measured for both the devices in the NV-RS mode and V-RS mode. Error bar: resistance distribution obtained from three randomly selected devices. Cross-sectional TEM images of the TST device after (**b**) NV-SET switching and (**c**) V-TS.

COMSOL simulations are carried out to study the different temperature fields evolved in a Te filament-based device and a Ag filament-based device under the same CC of 1 mA. According to our TEM analysis, truncated-cone-shaped filaments are adopted in the simulations. Both systems under simulations adopt the same geometry and size. The results of the simulations are depicted in Supplementary Fig. S9, from which we find that heat is more localized at the thinner end of the Te filament than is in the Ag filament because Te has poorer thermal conductivity. In addition to the stronger thermal confinement for the Te filament observed from the simulations, it is also clear that under the same simulated CC of 1 mA the highest local temperature in the Te filament reaches its melting temperature, while the melting temperature of Ag (961 °C) is not reached in the Ag filament-based device.

Based on the experimental and simulation results, a comparison between the operation mechanism of the Te filament-based device and that of the conventional metallic filament-based device is schematically depicted in Fig. 3. For the Te filament-based device in its initial HRS (Fig. 3a1), the NV-RS process occurs if a sufficiently negative voltage is applied to the top Te active electrode (Fig. 3a2). The switching process involves the following steps:

(i) cathodic dissolution of Te according to the reaction $Te+2e^- \rightarrow Te^{2-}$;
(ii) drift of $Te^{2-}$ anions across the dielectric thin film under the action of the high electric field;
(iii) oxidation of $Te^{2-}$ and electro-deposition of Te on the surface of the counter electrode according to the reaction $Te^{2-} \rightarrow Te+2e^-$.

The chemical state of Te ions involved in the electrochemical reactions is believed to be $Te^{2-}$ which is the formal reduction state of Te. The ability to be reduced distinguishes Te from other common electrochemically active metals, such as Ag and Cu. A key evidence of the occurrence of the electrochemical reduction has been provided in a previous work[39] where the SET switching was observed to occur only when negative voltage bias was applied to the Te electrode in the asymmetric Te/dielectric/Pt device.

The electro-deposition process (iii) leads to the growth of a Te filament. After the Te filament has grown sufficiently long to short-circuit the two electrodes, the cell is switched to the LRS (Fig. 3a3).

The cell retains the LRS unless a sufficiently large voltage of opposite polarity is applied and the electrochemical dissolution of the Te filament RESETs the cell to its initial HRS. During RESET

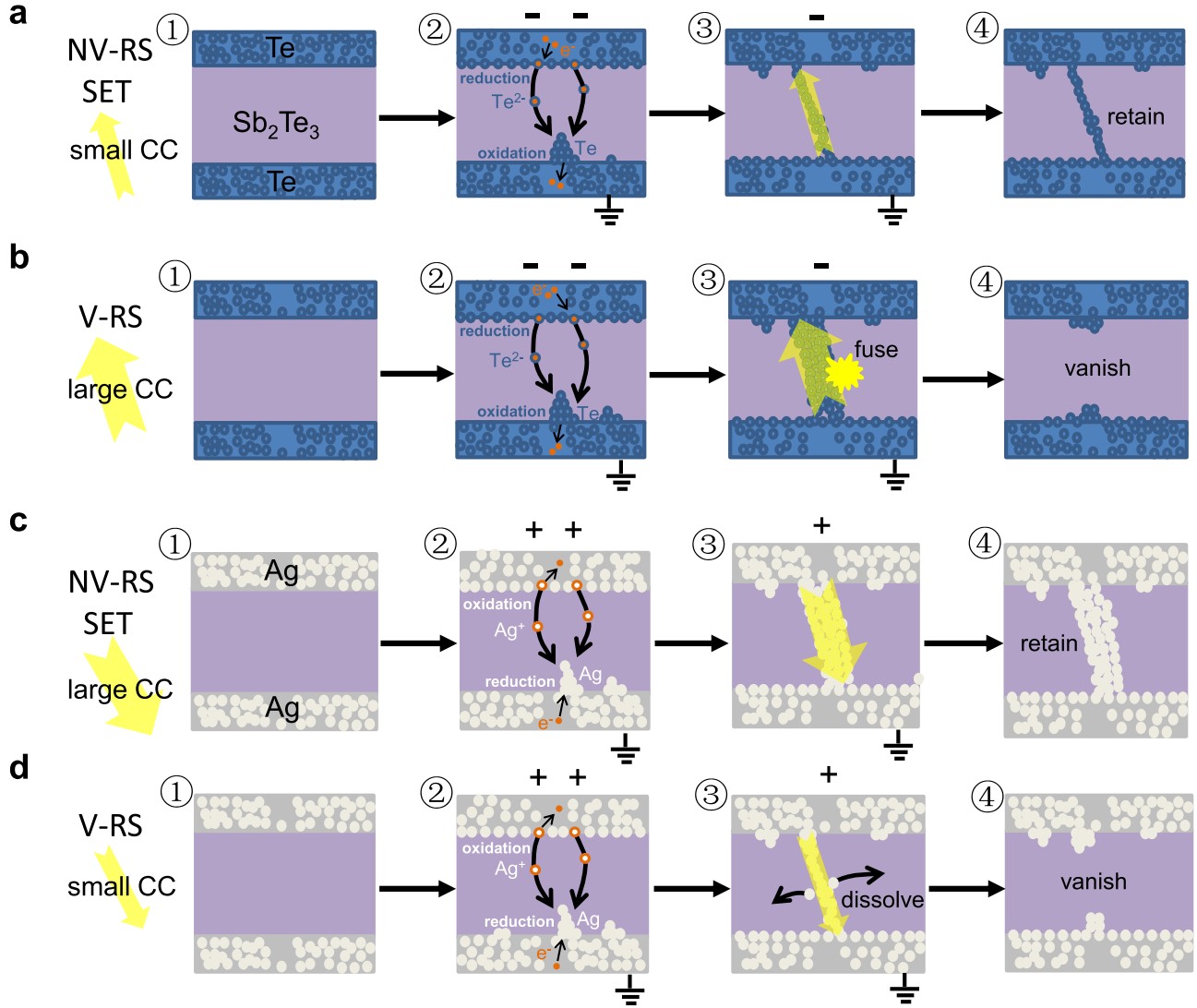

**Fig. 3 Schematic comparison of the filamentary RS processes between the Te filament-based device and the metallic filament-based device.** (**a**) NV-RS process under low CC in TST device. (**b**) V-RS process under large CC in TST device. (**c**) NV-RS process under large CC in Ag filament-based device. (**d**) V-RS process under low CC in Ag filament-based device.

(as shown in Supplementary Fig. S10), the Te filament is dissolved by reversing the electrochemical processes:

(i) dissolution of the Te filament at its end near the bottom electrode according to the reaction $Te + 2e^- \rightarrow Te^{2-}$;
(ii) drift of $Te^{2-}$ anions across the dielectric thin film under the action of the high electric field;
(iii) oxidation of $Te^{2-}$ on the surface of the top electrode according to the reaction $Te^{2-} \rightarrow Te + 2e^-$.

Note that in the initial phase of RESET there has already been current through the Te filament. The generated Joule heat assists the out-diffusion and drift of the dissolved $Te^{2-}$ anions. The rupture of the Te filament at one end or at its weakest point (process i) results in sudden increase of the device resistance and consequently sharp decrease of the current. This terminates the rupture process and leaves the main body of the filament intact. Therefore, the successive SET switching may only be a matter of restoring the tiny ruptured point instead of reconstructing the whole filament.

To ensure that the Te-filament-based device can be operated in the NV-RS mode, a low enough compliance current must be set so that

the current passing through the growing Te filament will not be too high to fuse it. However, if high current is allowed to pass through (Fig. 3b3), the accumulated heat becomes sufficient to fuse the just-grown Te filament with low-melting temperature (Fig. 3b4). In this case, the operating mode of the device is transitioned to the V-RS.

For conventional Ag filament-based device, the basic steps involved in the NV-RS process are similar to those for the Te filament-based device except that opposite voltage polarity is required for anodic dissolution of Ag to $Ag^+$ cations. Because Ag has much higher melting temperature than Te, high current is helpful (as long as hard breakdown does not occur) for the formation of filament because the lateral diffusion of the Ag particles and therefore the thickening of the filament is facilitated at the elevated temperature by the JH effect (Fig. 3c3). On the contrary, if the current is low, the filament formed can be weak and unstable. Therefore, when the voltage ceases, the filament may be spontaneously ruptured, leading to V-RS (Fig. 3d4).

Before closing this section, we would like to point out that our TST device is a forming-free device. As commonly known, forming is a process through which sufficient mobile ions are generated in the dielectric for the subsequent filamentary

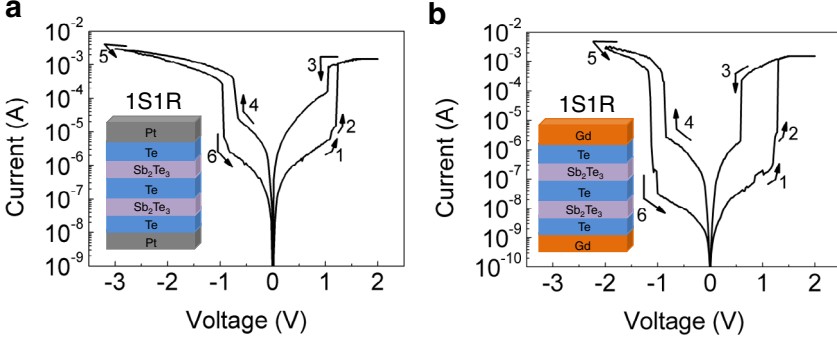

**Fig. 4 1S1R characteristics in a tandem of two identical TST devices.** *I–V* curves of the (**a**) PTSTSTP device and (**b**) GTSTSTG device.

switching. Sufficient amount of $Te^{2-}$ ions already exists in the $Sb_2Te_3$ dielectric and therefore additional forming is not required.

**One-selector–one-resistor function**. Because the Te-based RS device has the desired selector and memory properties which can be selectively used in the respective operating modes, a natural question arises here is whether Te-based RS device has the potential to be a universal RS cross-point array building block. To investigate, we fabricate the Te/$Sb_2Te_3$/Te/$Sb_2Te_3$/Te (TSTST) structure that is composed of two identical (in the sense that the fabrication conditions and the physical dimensions are the same) TST cells in tandem. Quasi-DC voltage sweep (from 0 V) measurements are performed on this TSTST device. As seen from Fig. 4a, the current undergoes obvious increase twice during the forward voltage sweep (arrows 1 and 2). After the second current jump, the current reaches its maximum set by the 1.5-mA CC. These can be understood as the typical 1S1R phenomena. To be specific, even though the two TST cells in tandem have intentionally been fabricated to be the same, in practice it is simply not possible for them to be precisely the same. This inherent device-to-device variation results in slightly faster switching-on of one device which then becomes the selector device as in the usual 1S1R structure. The delayed one becomes the memory device naturally. The former and latter devices are responsible for the first and second current jump, respectively.

A main functional difference between our TSTST structure and the usual 1S1R structure is that in the former structure the selector or memory device is probabilistically determined but in the latter structure, they are deterministically designated. Therefore, we would like to call the selector and memory devices in our TSTST structure the lucky selector (l-selector) and lucky memory (l-memory).

During the backward voltage sweep, a sharp decrease of the current is observed before the voltage has become zero. This is another typical 1S1R phenomenon that indicates the turning-off of the selector device. Admittedly, we are currently not able to verify experimentally which device is turned off, it is very likely that it is the l-selector being turned off because it has been switched on earlier and heat accumulation in it is more pronounced.

After that, the $\log|I|–V$ curve retains the characteristic of high resistance (selector-off) till the sweeping voltage changes polarity. When the voltage becomes sufficiently negative, abrupt increase of current occurs once again. This is also a 1S1R phenomenon behind which is the switching-on of the bidirectional selector. If we further increase the negative voltage and then sweep it back, we notice a counter-clockwise hysteresis. This suggests that the memory has been reset to its HRS. Further decrease of the voltage gives rise to sharp current decrease, indicating that the selector

has also been turned off. Supplementary Fig. S11 sketches the principles of 1S1R operations of the TSTST device.

Similar 1S1R characteristics are also observed in tandem device with the Gd PEs. The on/off ratio is larger as expected, as seen from Fig. 4b. As far as we know, this is the first demonstration of the 1S1R behavior in tandem structure composed of two identical cells. It is worth noting that, from reliability consideration, practical applications may prefer deterministic selector and memory to lucky ones. In those cases, Te-based RS device can still be a promising candidate because determinacy can be engineered into the tandem structure by intentionally making the two component cells different, such as the use of different dielectric materials (as will lately been seen in Fig. 6) or simply different film thicknesses without changing the materials, which may still be far easier than the optimization of other conventional 1S1R devices, such as OTS: RRAM. The possibility to use a single set of materials for 1S1R application may simplify the design and save otherwise substantial materials optimization effort on two types of devices with distinct operating requirements.

**Long-term plasticity-to-short-term plasticity transition and low-pass filter application**. Another result of the adversarial EC-JH effect is the unusual long-term plasticity (LTP)-to-short-term plasticity (STP) transition under high-frequency pulse stimulations. In these pulse train measurements, the width and the amplitude of the pulse are fixed at 10 µs and 0.6 V, respectively, to avoid the over-accumulation of heat in a single pulse (Supplementary Fig. S5). The stimulus frequencies are adjusted by changing the pulse intervals. When the interval is set to 10 µs, an analog RS behavior is observed that the resistance of the device is gradually decreased, as shown in Fig. 5a. Interestingly, however, when the interval is reduced to 1 µs, the device first undergoes a more rapid decrease in resistance and lately a sudden resistance increase back to an early HRS (Fig. 5b), mimicking the transition from LTP to STP. This can be understood as the result of the accumulation of heat with the number of pulses that gradually surpasses the EC effect. High pulse frequencies prevent the heat from too much dissipation during the intervals.

This LTP-to-STP transition phenomenon is unusual in the sense that it contrasts with the commonly observed STP-to-LTP transition in metallic filament-based RS devices[58] which has been used for memory consolidation applications. Here, we propose an application of our device with unusual LTP-to-STP transition under high-frequency pulse stimuli as a temporal filter[59] in the event-based vision tasks. Event-based camera, based on dynamic vision sensor (DVS), is a kind of bio-inspired camera that senses continuous flows of asynchronous spatial events, and responds as they occur or stays silent otherwise, as shown in Fig. 5c. We use the event-based Neuromorphic MNIST (N-MNIST) dataset as

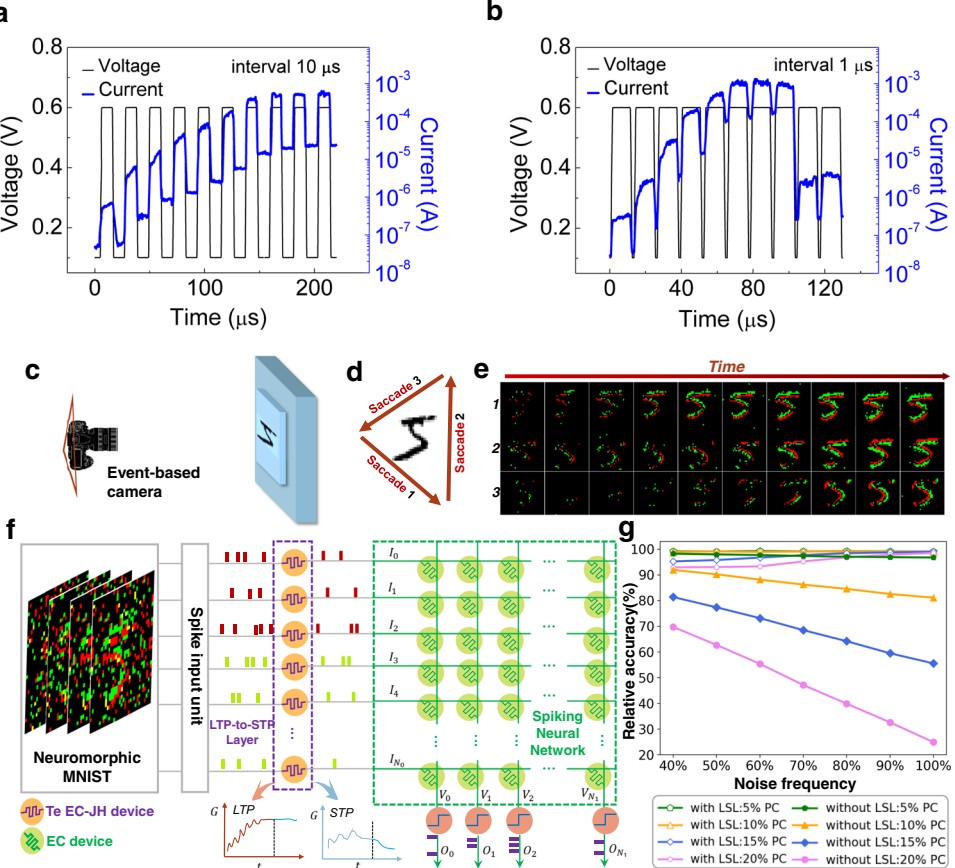

**Fig. 5 LTP-to-STP transition under pulse train stimuli and the proposed spike event preprocessing application.** (**a**) Pulse train measurement with write pulse width of 10 µs, amplitude of ±0.6 V and 10-µs interval. (**b**) Pulse train measurement with the same pulse width and amplitude, but 1-µs interval. (**c**) Schematic diagram of neuromorphic data acquisition via event-based camera. (**d**) Three saccade trajectories of the camera during data acquisition process. (**e**) Event-based data acquired by the camera along the three saccade trajectories. (**f**) Schematic diagram of SNN processing of the N-MNIST data with LSL. (**g**) Performance comparison between SNNs with LSL and without LSL.

our training and test sets for a five-layer spiking neural network (SNN)[60]. Compared to traditional frame-based MNIST datasets, N-MNIST contains richer temporal features and sparser information representations. As shown in Fig. 5d, each original image frame from MNIST dataset can be transformed into dynamic events by the saccade of the event-based camera. Figure 5e represents the recording results for a sample image of digit 5 along different saccade paths. Two detection channels that respond to different brightness change directions of the pixels, turning brighter and darker, are shown in red and green, respectively.

Despite its bio-plausibility and low power consumption, the event-based camera usually suffers from noise disturbance[61]. To simulate the inherent noise during visual information acquisition, we use uniform distribution function to produce a certain noise pattern (indicated spatially by pixel coverage, or PC) and add the noise to a certain percentage (indicated temporally by noise frequency) of the N-MNIST test data. The event-based SNN visual recognition framework is depicted in Fig. 5f. The SNN has already been trained using noise-free training data from the same dataset. During test, a LTP-to-STP layer (LSL) before the SNN was added to preprocess the visual inputs. The LSL is parameterized according to the measured behavior of the real Te-based devices. In the LSL, event noises with high frequencies are filtered out because of the temporal filter function, or more specific, low-pass filter function of the LSL. Genuine event signals

remain almost intact due to their inherent low-frequency nature. After preprocessing, visual recognition continues as usual by the following SNN processing.

The recognition accuracy is shown in Fig. 5g. It is seen that with increasing noise frequency the performance of the SNN without LSL degrades monotonically. In particular, the relative recognition accuracy for highly noisy (100%) data with respect to that for noise-free data decreases to below 30%. With the LSL included, the problem of accuracy degradation is mitigated. Interestingly, with increasing noise frequency the recovery of the accuracy is more complete. This is understandable from the frequency-dependent degree of LTP-to-STP transition. For highly noisy (100%) data, the accuracy could even be recovered to the same level as noise-free data recognition. The simulation results provide a glimpse of the potentially new computational advantages of the Te-based RS device.

It is noted that, in the proposed application of our device in the preprocessing layer for event-based vision tasks, the device is not necessarily driven back-and-forth between its on and off states by each received event pulse, but rather gradually changes its resistance state by accumulating the stress from multiple pulses after which either a LRS or a HRS is reached in response to the genuine event signals or the persistent (high-frequency) noises, respectively. In this sense, the endurance requirement for the device in this specific application may be less stringent.

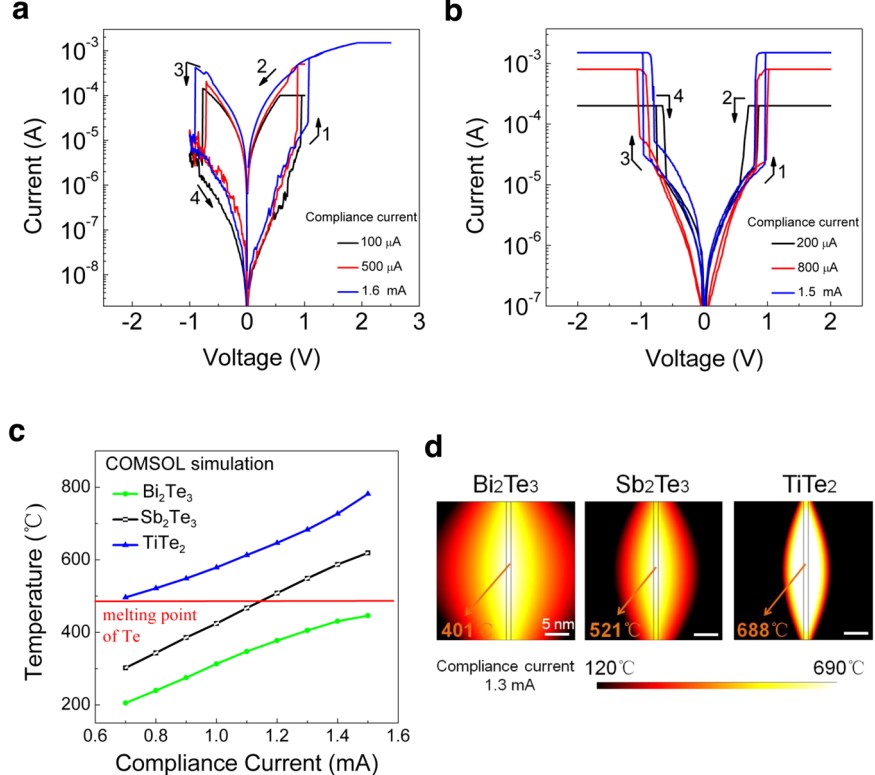

**Fig. 6 RS characteristics of the TBT and TT\*T control devices, and thermodynamics simulations of the temperature fields established during filamentary conductions.** I–V curves of the Pt protected (**a**) TBT device and (**b**) TT\*T device under various CCs. (**c**) Simulated CC-dependent local peak temperature in TBT, TST, and TT\*T devices. (**d**) Simulated temperature distributions in these three devices under 1.3-mA CC.

**Effects of the dielectric thermal conductivities.** In order to satisfy the device requirements in different application domains, the ability to regulate the RS behavior is important. Previous studies have shown that for the same filamentary materials the dielectric matrixes might have significant influence on the RS behavior[57,62–64]. The rate of the EC redox process, ion mobility and dielectric conductivity have been considered as the main regulators. Compared to conventional metallic filaments, Te filament has low thermal conductivity and low melting temperature. Here, we further investigate whether the thermal conductivity of the dielectric has an effect on the RS behavior of the Te filament-based device. To this end, two control devices, Te/Bi$_2$Te$_3$/Te (TBT) and Te/TiTe$_2$/Te (TT\*T), whose binary telluride dielectrics have thermal conductivities higher and lower, respectively, than that of Sb$_2$Te$_3$ (~1.2 W m$^{-1}$ K$^{-1}$ [65] for Bi$_2$Te$_3$, ~0.78 W m$^{-1}$ K$^{-1}$ [66,67] for Sb$_2$Te$_3$ and ~0.12 W m$^{-1}$ K$^{-1}$ [66,67] for TiTe$_2$, also see Supplementary Fig. S12) are also investigated.

As illustrated in Fig. 6a, b, the TBT device is always operated in the NV-RS mode over the CC range from 100 μA to 1.6 mA. On the other hand, the TT\*T device is always operated in the V-RS mode over the CC range from 200 μA to 1.5 mA. The chosen CC ranges here are comparable to that previously tested for the TST device which undergoes NV-RS-to-V-RS transition at a certain CC within the range. Therefore, full functional-range tunable Te filament-based devices, from always-NV RS, to NV-to-V transitionable RS, then to always-V RS, are achieved by using Bi$_2$Te$_3$, Sb$_2$Te$_3$, and TiTe$_2$ dielectric materials, respectively, with decreasing thermal conductivities. These rich dielectric-dependent RS phenomena in Te filament-based devices can be understood from the varying degrees of rivalry between the EC and JH effects: high-(low-) thermal conductivity dielectric facilitates (suppresses) heat dissipation and therefore makes the JH effect less (more) pronounced to counteract the EC effect.

Thermodynamics simulations of these three devices under the CCs ranging from 700 μA to 1.5 mA (in the vicinity of the NV-RS-to-V-RS transition point of the TST device) are performed to study the temperature distributions in these devices. For simplicity, cylinder-shaped filaments of the same sizes (1 nm in diameter, 30 nm in length) are modeled. As shown in Fig. 6c, the simulated highest achievable local temperature in TBT device is still lower than the melting temperature of Te (EC effect dominated), whereas the lowest local temperature in TT\*T device is already higher than the melting temperature of Te (adversarial JH effect dominated). For TST device with intermediate dielectric thermal conductivity, a crossover is observed, at which the NV-RS-to-V-RS transition occurs (balanced EC-JH effect). In practical memory and selector applications, the Te filament-based RS devices can be more reliably operated near their respective performance limits, i.e., lowest and highest achievable operating currents, than the conventional metallic filament-based RS devices in the sense that the undesired transition between NV-RS and V-RS is less likely to occur. This is because the more we push the limits for Te filament-based selectors and memories the more distant are they away from the NV-RS-to-V-RS transition point. On the contrary, due to the current-volatility dilemma, the limits for metallic filament-based selectors and memories are near the transition point where undesired transition between NV-RS and V-RS may occur. Although not in full quantitative agreement with the experiments due to the use of simplified models, the simulated temperature trends well explain the experimental RS behaviors of the three devices. Figure 6d shows the temperature fields in these three devices under the CC of 1.3 mA. Due to the lowest dielectric thermal conductivity, the TT\*T device shows the most confined temperature distribution around the filament and its peak local temperature of 688 °C is the highest among the three,

**Table 1 $\Delta w$ versus pulse interval.**

| $i$ | 10 | 9 | 8 | 7 | 6 | 5 | 4 | 3 | 2 | 1 | 0.1 |
|---|---|---|---|---|---|---|---|---|---|---|---|
| $\Delta w$ | 1.000 | 0.846 | 0.689 | 0.544 | 0.411 | 0.313 | 0.215 | 0.160 | 0.101 | 0.054 | 0.016 |

which is sufficient to fuse the Te filament. In contrast, the TBT device with the highest dielectric thermal conductivity shows the most expanded temperature distribution and its peak local temperature of 401 °C is the lowest among the three, lower than the melting temperature of Te. The TST device is intermediate.

To conclude, we demonstrate a new application opportunity for Te, that is, making RS devices. Te-based RS devices solve the long-standing current-volatility dilemma that has prevented the drive currents of selector devices from going higher and the switching currents of memory devices from going lower. Our proof-of-concept Te/Sb$_2$Te$_3$/Te device ($2 \times 2$ μm$^2$) can be operated in the NV-RS mode under CC around several μA as well as in the V-RS mode under CC around a few mA. These phenomena can be attributed to several indispensable materials properties combined in Te, namely, semiconductivity, electrochemical redox ability, low thermal conductivity, and low melting point, which can give rise to adversarial EC and JH effects to reverse the usual current dependence of filament stability. We also observe the 1S1R behavior in a tandem of two identical Te-based RS devices, demonstrating the potential of Te-based device as a universal building block for the RS cross-point array. The degree of rivalry between the EC and JH effects can be varied by the frequency of pulse stimuli, leading to unusual LTP-to-STP transition at high pulse frequency that can be considered as a useful computational source. A combination of unique electrical-thermal properties makes Te an attractive and promising enabler for future RS devices with large and unique design space.

## Methods

**Device fabrication.** Electrode/dielectric/electrode-structured cross-point devices with various junction areas ($2 \times 2$ μm$^2$, $4 \times 4$ μm$^2$, $8 \times 8$ μm$^2$, $16 \times 16$ μm$^2$) are fabricated on a thermally oxidized Si substrate. For the Pt-protected TST device, a 50-nm-thick Te bottom electrode is deposited via radio frequency (RF) sputtering directly on a 50-nm-thick Pt adhesion layer (via DC sputtering) beneath it. For the Gd protected device, a 30-nm-thick Gd layer is deposited before the deposition of Te. Photo-lithographically patterned Sb$_2$Te$_3$/Te/Pt (Sb$_2$Te$_3$/Te/Gd/Pt) stacked films are then deposited, completing the fabrication of a Pt (Gd) protected device. The Sb$_2$Te$_3$ film is prepared by RF sputtering from a stoichiometric target.

Two other dielectrics, Bi$_2$Te$_3$ and TiTe$_2$, are used in control devices. The TBT and TT*T devices are fabricated by the same process as that for the TST device.

The bottom protective electrodes of diameters of 60 and 150 nm for the T-shape T′TSTT′ devices are patterned by electron beam lithography. The $50 \times 50$-μm$^2$ dielectric layers and top electrodes are then patterned by photolithography on the pillar TiN bottom protective electrodes.

**Characterization.** The cross-section TEM specimens are prepared by focused ion beam (FIB) technique in Field Emission-Environment Scanning Electron Microscope (QUANTA 200 FEG). 2 keV Ga ion beam is used to cut the cell. The HRTEM images and EDS elemental analysis are accomplished with a Field Emission Gun/TEM (JEM-2100F) operated under 200 kV voltage.

**Electrical measurement.** Cyclic quasi-DC voltage sweep measurements are performed by the Keysight B1500A semiconductor analysis system. The Keysight B1530A waveform generator/fast measurement unit is used to perform the pulse measurements. Using a two-probe (W tips) configuration, DC and pulsed voltages are applied to one electrode with the other electrode grounded.

LinKam T96-S heating-stage is used for heating devices in the temperature-dependent conductivity measurements.

**COMSOL thermodynamics simulation.** The filament is modeled as a structure embedded in the dielectric matrix and connecting the two electrodes at its two ends. In numerical simulations, the filament is supposed to be a large number of plates stacked in series. Each of the plates can be approximated to cylinder shape as long as the thickness $dz$ is small enough. The resistance $R(z)$ of the plate can be described as,

$$R(z) = \frac{dz}{\sigma(T)S} \quad (1)$$

where $S$ is the top or bottom surface area of the cylindrical plate and $\sigma(T)$ is the temperature-dependent electrical conductance, which can be expressed as,

$$\sigma(T) = \frac{\sigma_0}{1 + \alpha(T - T_0)} \quad (2)$$

where $T_0$ is the room temperature, $\sigma_0$ is the electrical conductivity of the filament material at room temperature and $\alpha$ is the temperature coefficient of resistivity. According to Joule's law and the Fourier heat equation, the global thermal behavior can now be described as,

$$\begin{cases} d_z\rho c_p S \cdot \nabla T + \nabla \cdot \mathbf{q} = d_z Q + h \cdot (T_{\text{ext}} - T) \\ \mathbf{q} = -d_z k \nabla T \\ Q = I^2 R t \end{cases} \quad (3)$$

where $\rho$ (Te: 6250 kg/m$^3$; Ag: 10,490 kg/m$^3$), $c_p$ (Te: 164 J kg$^{-1}$ K$^{-1}$; Ag: 235 J kg$^{-1}$ K$^{-1}$), $h$ (Bi$_2$Te$_3$: 1.2 W m$^{-1}$ K$^{-1}$; Sb$_2$Te$_3$: 0.78 W m$^{-1}$ K$^{-1}$; TiTe$_2$: 0.12 W m$^{-1}$ K$^{-1}$), and $k$ (Te: 1.6 W m$^{-1}$ K$^{-1}$; Ag: 429 W m$^{-1}$ K$^{-1}$) are density, heat capacity, external heat transfer coefficient, and thermal conductivity of the filament, respectively. The filament-dielectric and the filament-electrode thermal exchange are taken into account. They are different in the external heat transfer coefficient because the thermal conductivities of the electrode and dielectric are different. Because the temperature dependence of $k$ is much weaker as compared to that of the electric conductance, we regard it as constant in the simulations.

**SNN training.** The leaky integrate-and-fire (LIF) model as the basic neuron unit is adopted and spatial-temporal backpropagation (STBP) algorithm[68,69] for training is used. K3S1P1C128- K3S1P1C256- K3S1P1C256- FC128-FC10 (K: convolution kernel size, S: stride, P: padding, C: output channel, FC: full-connected dimension) network structure is used. Adam optimizer[70] with an initial learning rate of 0.0005 and dropout technique are used. The dropping proportion is set to 0.25 for first layer and 0.4 for the other layers. The network is pretrained using 50,000 training samples with a batch size of 100 for 50 epochs. 10,000 noisy test samples are used in the test procedure. The time window is set to 8, the threshold to 0.4, and the decay factor to 0.5. Gradient substitution method with rectangular length of 0.5[68,69] is used. All simulations are performed using PyTorch on one RTX 2080Ti GPU.

**LSL parameterization.** To parameterize the LSL, pulse trains each consisting of 10 pulses with identical amplitude (+0.6 V) and width (10 μs) but different intervals (0.1, 1, 2, 3, 4, 5, 6, 7, 8, 9, and 10 μs) are applied to the TST devices. We record the conductances of the devices before and after the pulse train stimulations. The differences of the conductances between the initial and final states are then normalized as values of weight changes ($\Delta w$), as seen in Table 1. Finally, the $\Delta w$-interval relationship is extracted by exponential fitting. The fitted curve is $w = 0.205 \cdot e^{\frac{i}{5.549}} - 0.192$, where the interval is denoted as $i$.

## Data availability

All data needed to evaluate the conclusions in the paper are present in the paper and/or the Supplementary Materials. Additional data related to this paper is available from the authors upon reasonable request. Source data are provided with this paper.

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

## Acknowledgements

The authors acknowledge funding from National Natural Science Foundation (grant nos. 61974082, 61704096, and 61836004). The authors acknowledge funding from National Key R&D Program of China (2018YFE0200200), Youth Elite Scientist Sponsorship (YESS) Program of China Association for Science and Technology (CAST) (no. 2019QNRC001), supercomputing wales project number scw1070, Tsinghua-IDG/McGovern Brain-X program, Beijing science and technology program (grant nos. Z181100001518006 and Z191100007519009), the Suzhou-Tsinghua innovation leading program 2016SZ0102, and CETC Haikang Group-Brain Inspired Computing Joint Research Center.

## Author contributions

H.L. conceived the idea and supervised the project. Y.Y. and X.W. fabricated the μm-scale devices. S.J. and M.Z. fabricated the nm-scale devices. Y.Y. and S.J. conducted the device measurements and characterizations. B.W. and K.L. assisted the electrical measurements. L.X. assisted the device fabrications under the supervision of W.D. H. Liu and Y.L. assisted the characterizations under the supervision of D.L. M.X. conducted the SNN simulations under the supervision of J.P. Y.Y., X.W., and Y.G. conducted the COMSOL simulations. Y.Y. and H.L. wrote this paper. All the authors discussed the results and commented on the manuscript.

## Competing interests

The authors declare no competing interests.
