## [Peer Review File · Nature Communications]

Reviewers' Comments:

Reviewer #1:

Remarks to the Author:

This article reports a type of tellurium semiconductor based resistive switching device, where a Te filament is proposed to be responsible for the switching mechanism. By manipulating the thermal conductivity of the filament and dielectric as well as input parameters such as current compliance (CC), pulse amplitude and pulse interval, the device can be controlled to exhibit nonvolatile switching under low CC while volatile switching under high CC, therefore addressing the current-volatility dilemma. The proposed concept is potentially interesting, whereas there are still some major concerns related to the mechanism and performance of the device, which should be addressed by the authors.

1. If Te is electrochemically active and the filament is composed of Te, what is the chemical state of Te ions involved in the electrochemical reactions? The authors need to provide a complete physicochemical picture for the switching process.
2. The application of the device as a preprocessing layer is actually using it as a low-pass filter based on the LTP-STP transition upon high-frequency pulse stimulations. Such application necessitates high endurance of the device under input pulses. However, the endurance of the device is too low to be applicable, which are 100 and 50 times for the nonvolatile and volatile operation modes, respectively.
3. The entire logic of the work is based upon the assumption that Te filament(s) is responsible for the resistive switching in the device, and therefore it is the backbone of the study. However, the TEM observation shown in Fig. 2 does not seem convincing, because the filament region is not clearly defined and observations were not made before and after switching to identify the exact region responsible for the switching mechanism. I recommend further clarifying the switching mechanism, due to its importance for the validity of the assumption.
4. The switching characteristics of the proposed devices may limit their actual applications. For example, 1) the nonvolatile switching takes place in a CC range of 0.3-0.9 mA. Such operation current is too high, and it doesn't seem to have a margin for optimization, because even the off state of the device is very conductive, as shown in Fig. 1; 2) as limited by the highly conductive off state, the on/off ratio of the device, especially in the nonvolatile mode, is too small (Fig. 1c); 3) besides the different CC values used, the off state in the nonvolatile (Fig. 1c-d) and volatile (Fig. 1e-f) operations also seems very different. Will this contribute to the different switching behaviors as well? 4) the current density of 6×10^4 A/cm² in the volatile mode compares unfavorably with those already reported in the literature.
5. Does the device require a forming process before set/reset? I found this never mentioned in the manuscript.
6. The abbreviation "LSL" should be defined when it is used for the first time.
7. The references might be too much, and the numbering in the text seems to be discontinuous.

Reviewer #2:

Remarks to the Author:

The author's manuscript shows that resistive memory switching and threshold switching are at low and high current levels, respectively, due to the low thermal conductivity and low melting point of Te in the same device.

This manuscript shows its originality and novelty as following. It is explained that it works (solves the current-retention (or current-volatility) dilemma in a resistive switching device), and this result is applied to LTM-to-STM. In the reference [56] presented by the author as an example of solving the current-retention dilemma, memory and threshold switching are performed at the 1 μ A and 1mA levels, respectively. In order to operate, the graphene of the interface was formed differently.

On the other hand, the results of this study show that without graphene, which is difficult to directly apply to CMOS devices, It is explained that it has high applicability because it solves the dilemma by using the same Te-based device.

To be accepted, I strongly recommend authors explain sufficiently following questions and comments

1. In this manuscript, like the graphene control in [56], a key factor that can clearly distinguish between memory and threshold needs to be presented.
2. The DC memory operation window in Figure 1 is narrow. It does not seem a stable memory operation, and seems to be a temporary memory effect incidental to DC operation.
3. Even in the explanation of the operation principle in Fig.3, the memory operation is not explicitly explained.
4. In the LTM-to-STM application, threshold switching is not focused on the application by solving the memory-threshold dilemma. Only the applicability used as a high frequency noise filter is presented. I recommend explain more detail.
5. Since the temperature distribution in Fig. 6 explains the contention between memory-threshold operations, it can be explained that Sb₂Te₃ is in the transition period of the two switching modes. However, it is understood that it is difficult to operate the two operations independently. (Related to comment #1)
6. Memory On/Off Ratio is too small to support memory operation, which is critical for real array operation.

Reviewer #3:

Remarks to the Author:

The authors proposed an interesting approach to overcome the current-volatility dilemma of electrochemical resistive switching (RS) devices by utilizing a Te filament in a Te/Sb₂Te₃/Te structure. Semiconducting Te has lower thermal conductivity and lower melting temperature than metal materials (such as Cu and Ag) usually adopted in electrochemical RS devices. Owing to the stronger thermal confinement and higher local temperature around the Te filament, the device exhibits non-volatile RS (NV-RS) with low ON current and volatile RS (V-RS) with high ON current, which fulfill the requirements for memories and selectors in cross-point arrays. The authors also demonstrate unusual stimulus frequency dependent long-term memory to short-term memory transition under voltage pulse trains and tunable switching behaviors between NV-RS and V-RS by changing the telluride matrix. The paper is well written and contains new results of reasonable significance. However, I would like to have the authors to consider the following issues and respond to them.

- 1) Throughout the paper, the style of figures is incorrect and confusing. The alphabetical characters (a, b, c, etc.) in the bottom left-hand side corner should be moved to the upper left-side corner.
- 2) The elemental mapping of TEM image (Fig. 1b) is difficult to recognize where the Te/Sb₂Te₃ and Sb₂Te₃/Te interfaces are located. Why the authors show the images with different magnifications in Figs. 1a and 1b?
- 3) In NV-RS shown in Figs. 1c and 1d, the current compliance is always set to a higher value in

the negative bias than in the positive bias. What is the role of the current compliance in the negative bias? Is it necessary to realize the NV-RS behavior?

4) In the simulations of the temperature distribution, as shown in Figs. 2d and 2e, there is no information about the filament size. The authors should describe the diameter on both side and the filament length in the text. The scale bar should also be indicated in the figures.

5) Although the authors discussed the difference between Ag filament-based and Te filament-based devices in Fig. 2d and 2e, the explanation of Fig. 3 is not enough. I think the authors should discuss the difference in switching mechanism between Ag filament-based and Te filament-based RS memories in more detail using Fig. 3. Otherwise, Fig. 3 should be moved to Supplementary Information.

6) On page 9, line 3: '1 μ s' should be '10 μ s'.

7) The endurance results under voltage pulse trains (Figs. 4c and 4d) show that the ON and OFF resistances decrease with increasing the number of voltage pulses. These behaviors imply that the degradation of the device performance takes place in the pulse train measurements even with 1 μ s pulse width. How many cycles can the device repeat RS? The endurance data for more cycles had better put in Supplementary Information.

8) In the simulations of the temperature distribution shown in Fig. 6d, the diameter and length of the filament should be described. Is the thermal conductivity of the Te electrode on both sides taken into account (also for the results of Figs. 2d and 2e)? The detailed conditions in the simulations may be better to described in the Experimental section or in Supplementary Information.

We thank you very much for your valuable time and useful suggestions for improving the quality of the manuscript. After carefully considering your comments, we have carried out additional experiments and revised the manuscript substantially to address all the reviewers' comments. The text changes are shown in blue.

In the original manuscript, we reported that the Te filament-based resistive switching (RS) devices can **novelly solve the long-lasting current-volatility dilemma** that has been widely faced by various resistive switching device implementations and has prevented the operating currents of selector and memory from going higher and lower, respectively, as desired. **This is related to a combination of unique electrical-thermal properties in Te, including its semiconductivity, electrochemical redox activity, low thermal conductivity and low melting temperature.**

In addressing your comments, we have tapped even more potentials of this device, further indicating the promise held by Te filament-based RS devices. These main device properties include:

1. Switching current as low as 25 μA (5 μA) for Pt (Gd) protected Te/Sb₂Te₃/Te device in the NV-RS mode.

These switching current values are among the lowest for electrochemical metallization RS devices. In particular, they are lower than that (30 μA) of the state-of-the-art Ag filament-based RS device with substantial dielectric optimization^[1] and comparable to that (1 μA) of the recently reported rare-earth Ru filament-based RS device.^[2] **This can be understood from the relatively low conductivity of the semiconducting Te filament compared to metallic filaments.** We anticipate to further push the current limit by dielectric optimization, e.g., using oxide dielectrics, which is the scope of the next work.

2. Estimated some tens of MA/cm² current density in the V-RS mode if the device could be further scaled down to the nanometer scale.

In response to one of the reviewers' comment, we have realized that the originally presented several mA drive current in our 2x2 μm^2 device operating in the V-RS mode suggested the likelihood to achieve some tens of MA/cm² current density if the device could be further scaled down to the nanometer scale. **This is based on the hypothesis that the drive current is unchanged considering its filamentary origin. If so, this drive current density could rival that of the state-of-the-art nanoscale OTS selector.**

3. Nearly 100 times increase of the on/off ratio by protective electrode (PE) optimization, i.e., replace the high-work function (WF) Pt PE (5.9 eV) with low WF Gd PE (2.9 eV).

Again, this is related to the semiconducting property of Te. To be specific, it is known that Te is generally p-type and therefore the use of low WF Gd PE increases the Schottky barrier height at the PE/Te interface and consequently the contact resistance. **As far as we know, this is the first report on the PE electrode effect on RS device performance and the exploitation of such effect.**

4. 1S1R behavior in tandem device that is composed of two identical Te/Sb₂Te₃/Te cells.

As far as we known, this is the first demonstration of 1S1R behavior in a tandem of two identical component cells. This is in stark contrast to any of the previously reported 1S1R

phenomena, which requires two different cells out of different materials and with different RS mechanisms for NV-RS memory (e.g., RRAM or PCRAM) and V-RS selector (e.g., Mott or OTS), respectively. **This indicates the potential of Te-based device as a universal RS array building block.** The possibility to use a single set of materials for 1S1R application may simplify the design and save otherwise substantial materials optimization effort on two types of devices with distinct operating requirements.

Point-by-point response to reviewers' comments

Reviewer #1

General comment

This article reports a type of tellurium semiconductor based resistive switching device, where a Te filament is proposed to be responsible for the switching mechanism. By manipulating the thermal conductivity of the filament and dielectric as well as input parameters such as current compliance (CC), pulse amplitude and pulse interval, the device can be controlled to exhibit nonvolatile switching under low CC while volatile switching under high CC, therefore addressing the current-volatility dilemma. The proposed concept is potentially interesting, whereas there are still some major concerns related to the mechanism and performance of the device, which should be addressed by the authors.

Response

Thank you very much for finding our work of potential interest and giving us constructive comments to further improve the quality of this work. According to your suggestions, we have fabricated new devices and carried out more thorough device characterizations and optimizations.

Our responses to your specific comments one by one are shown as follows.

Comment 1

If Te is electrochemically active and the filament is composed of Te, what is the chemical state of Te ions involved in the electrochemical reactions? The authors need to provide a complete physicochemical picture for the switching process.

Response

We would like to thank the reviewer for pointing out this issue that is important for the readers to better understand the resistive switching mechanism of our device. The chemical state of Te ions involved in the electrochemical reactions is believed to be Te^{2-} which is the formal reduction state of Te. The ability to be reduced distinguishes Te from other common electrochemically active metals, such as Ag and Cu. In our previous work, a key evidence of the occurrence of the electrochemical reduction has been provided,^[3] that is, SET switching occurred only when negative voltage bias was applied to the Te electrode in the asymmetric Te/insulator/Pt device.

In the revised manuscript, we have added discussions on the chemical state of Te ions involved in the electrochemical reactions and presented it schematically in **figure 3**. On **page 10**, the text revision reads “The chemical state of Te ions involved in the electrochemical reactions is believed to be Te^{2-} which is the formal reduction state of Te. The ability to be reduced distinguishes Te from other common electrochemically active metals, such as Ag and Cu. A key evidence of the occurrence of the electrochemical reduction has been provided in a previous work where the SET switching was observed to occur only when negative voltage bias was applied to the Te electrode in the asymmetric Te/dielectric/Pt device.”

Comment 2

The application of the device as a preprocessing layer is actually using it as a low-pass filter based on the LTP-STP transition upon high-frequency pulse stimulations. Such application necessitates high endurance of the device under input pulses. However, the endurance of the device is too low to be applicable, which are 100 and 50 times for the nonvolatile and volatile operation modes, respectively.

Response

Thank you very much for your comment on this issue. We could not agree more that real-world applications require device optimizations in various aspects, including its endurance. **What we would like to clarify here is that the results of the 100 and 50 times of operations presented in the original manuscript were obtained under successive quasi-DC sweepings which applied much stronger stress to the device than did the pulses, and we had not intended to push the limit at that time.**

In the revised manuscript, we have further increased the number of the testing quasi-DC sweeping cycles to 200 in both operation modes of our device, i.e., high-current V-RS selector and low-current NV-RS memory. In principle, even larger number of cycles of quasi-DC sweeps can be applied, especially when the device is operated under the NV-RS memory mode because the switching current is lower. **As a comparison, Zhao et al^[4] reported I-V curves obtained by 100 times of quasi-DC sweepings.** To avoid misunderstanding and for better structured manuscript, we have presented these results in the supplementary information (**figure S3**). **It is also interesting to find that the replacement of the Pt protective electrode (PE) with Gd PE, which has originally aimed to increase the dynamic range of the RS (in response to comment 4), enhances the endurance performance (figure S3 and S4). It is likely that the better performance of Gd protected device originates from the smaller electronegativity of Gd (1.21) compared to Pt (2.28) and the consequent stronger binding to Te. This may suppress the injection of excessive Te²⁻ into the dielectric and therefore mitigate filament overgrowth which is generally believed to be a main culprit for the endurance issue.^[5]**

Results of the pulse train measurements have also been given in **figure S4** of the revised manuscript for both Pt and Gd protected devices, respectively. It is seen that while the state of low resistance (LR) is reasonably stable, the state of high resistance (HR) suffers from certain degree of degradation with increasing number of stressing pulses. This looks more severe for the device operated in the V-RS mode, which can be understood from the more intense stress applied. Similar phenomena have also been frequently observed in conventional Ag and Cu-based ECM cells and have generally been attributed to filament overgrowth.

We have updated the texts regarding the quasi-DC endurance tests and pulse endurance tests, and have added discussions on the performance difference between the Pt and Gd protected devices in the revised manuscript. On **page 8**, the text revision reads “**Endurance tests have also been performed for both PTSTP and GTSTG devices in their respective NV-RS and V-RS operating modes, as shown in supplementary figure S3 and S4. Continuous quasi-DC sweeps and pulse train measurements show certain degree of on/off ratio degradation. Similar phenomena have also been frequently observed in conventional Ag and Cu-based ECM cells and have generally been attributed to filament overgrowth. In this respect, it is interesting to find that**

GTSTG device has better endurance performance than PTSTP device. It is likely that the better performance of Gd protected device originates from the smaller electronegativity of Gd (1.21) compared to Pt (2.28) and the consequent stronger binding to Te. This may suppress the injection of excessive Te^{2-} into the dielectric and therefore mitigate filament overgrowth.”

It should be pointed out that in the proposed application of our device in the preprocessing layer for event-based vision tasks, **the device is not necessarily driven back-and-forth between its on and off states by each received event pulse, but rather gradually changes its resistance state by accumulating the stress from multiple pulses** after which either a low-resistance state or a high-resistance state is reached in response to the genuine event signals or the persistent (high-frequency) noises, respectively. In this sense, the operating requirement for the device in this specific application is not as stringent as it appears to be.

We have added explanations about this issue in the revised manuscript. On **page 16**, the text revision reads “It is noted that, in the proposed application of our device in the preprocessing layer for event-based vision tasks, the device is not necessarily driven back-and-forth between its on and off states by each received event pulse, but rather gradually changes its resistance state by accumulating the stress from multiple pulses after which either a LRS or a HRS is reached in response to the genuine event signals or the persistent (high-frequency) noises, respectively. In this sense, the endurance requirement for the device in this specific application may be less stringent.”

Of course, how to optimize the endurance performance of our Te-based RS device is an important issue remains to be investigated in future works and **there are at least two possible solutions, scaling down the device and optimizing the dielectric layer**. Currently, our device is $2 \times 2 \mu\text{m}^2$. Scaling it further down will limit Te supply and confine Te^{2-} injection into the dielectric layer which may improve its endurance.^[6] In addition to spatial confinement of Te^{2-} injection, optimizing the dielectric layer by using less Te-dissolvable dielectric or inserting appropriate Te^{2-} ion buffer layer^[5,7-9] is also considered as a research direction worth pursuing, not only for its technological relevance but also for its fundamental scientific interest considering Te as an electrochemically active group VI elemental semiconductor.

In the revised manuscript, we have added discussions on this issue in the supplementary **note S1** after figure S4.

Te-based device technologies, especially RS devices, are still in their infancies, and we admit that the endurance performance of our present proof-of-concept Te-based RS device cannot rival the state-of-the-art Ag or Cu-based ECM devices after years’ continual optimization, from a performance level similar to ours^[10] to today’s 10^8 .^[11] However, we optimistically regard our results as indications of research opportunities rather than limitations for Te-based RS devices, and we believe that the technological potential of Te-based RS devices, given the demonstrated functions of novelty, will not be compromised.

Comment 3

The entire logic of the work is based upon the assumption that Te filament(s) is responsible for the resistive switching in the device, and therefore it is the backbone of the study. However, the TEM observation shown in Fig. 2 does not seem convincing, because the filament region is not

clearly defined and observations were not made before and after switching to identify the exact region responsible for the switching mechanism. I recommend further clarifying the switching mechanism, due to its importance for the validity of the assumption.

Response

Thank you very much for your comment. As you have pointed out, it could be more convincing if the characterization had been conducted in-situ. However, both practical and fundamental difficulties arise in the attempt to foresee the position in the pristine device from which the filament will form during the following SET switching. First, our device is $2 \times 2 \mu\text{m}^2$ in size and it is perpendicular. Therefore, the position of the formed filament can only be identified by extensive examinations of the SET sample.^[12] In the revised manuscript, we have carried out extensive examinations of several $2 \mu\text{m}$ -wide focused ion beam samples prepared from the as-deposited pristine devices. No filament-like structure has been found. Several randomly selected TEM images have been provided in the supplementary information (**figure S5**). Admittedly, the performed characterizations are not exhaustive. However, **the several facts, including the absence of any filament-like feature in our extensively searched cross-sectional areas in the pristine devices, the electrode size-independent LR (figure 2a) and the lack of other conceivable (semi)conducting filament compositions than Te, still strongly support the Te filament switching mechanism of our device.**

The reviewer's comment has indeed inspired us to study the electrochemical dynamics of nanoscale Te inclusions in various dielectrics with different Te^{2-} (or Te) diffusivity in future works by fabricating lateral RS devices and conducting in-situ characterizations. These studies may provide direct evidence of Te filament-based RS switching mechanism as well as scientific insights into the different electrochemical dynamics of semiconducting Te compared to metals at the nanoscale.^[13]

In the revised manuscript, we have added the supplementary results of the TEM characterizations and added the corresponding discussions in the supplementary **note S2** after figure S5.

Comment 4

The switching characteristics of the proposed devices may limit their actual applications. For example, 1) the nonvolatile switching takes place in a CC range of 0.3-0.9 mA. Such operation current is too high, and it doesn't seem to have a margin for optimization, because even the off state of the device is very conductive, as shown in Fig. 1; 2) as limited by the highly conductive off state, the on/off ratio of the device, especially in the nonvolatile mode, is too small (Fig. 1c); 3) besides the different CC values used, the off state in the nonvolatile (Fig. 1c-d) and volatile (Fig. 1e-f) operations also seems very different. Will this contribute to the different switching behaviors as well? 4) the current density of $6 \times 10^4 \text{ A/cm}^2$ in the volatile mode compares unfavorably with those already reported in the literature.

Response

Thank you very much for your comment which has inspired us to conduct performance limit test of our device and further explore its design space. As the reviewer has pointed out in

comment 4.1), our originally presented some hundred- μA SET switching currents for the NV-RS devices were high. In the revised manuscript, we have tried to push the limit. As shown in **figure 1c**, the SET switching current can be reduced to 25 μA . **This value is lower than that (30 μA) of the state-of-the-art Ag filament-based RS device with substantial dielectric optimization.**^[1]

Again, as the reviewer has mentioned, **the conductivity of the dielectric sets a fundamental limit on the lowest SET switching current that can be achieved. In this work, Sb_2Te_3 has been chosen as the dielectric in consideration of its thermal conductivity in between those of the other two control dielectrics, i.e., Bi_2Te_3 and TiTe_2 , to validate our assumption of the adversarial EC-JH in the design, as well as its desired composition, for instance no oxygen, to avoid the possibilities of forming filaments other than Te. Sb_2Te_3 is known to be a low-bandgap p-type semiconductor. Therefore, its conductivity is relatively high.** Investigating the use of wide-band oxide dielectric is a promising research direction in the future.

We have added this new result and further discussions in the revised manuscript. On **page 7**, the text revision reads “As previously introduced, low-operating current memory and high-ON current selector are desired. In this regard, our device is potentially superior in both of these two aspects. As shown in figure 1c, we demonstrate that the operating current limit of our device in the NV-RS mode can be pushed downward to 25 μA . This value is lower than that (30 μA) of the state-of-the-art Ag filament-based EC-RS memory device with substantial dielectric optimization. This can be understood from the relatively low conductivity of the semiconducting Te filament compared to metallic filaments.” On **page 7**, the text revision reads “As can be seen from figure 1c, our proof-of-concept device has relatively high OFF current which sets a fundamental limit on the lowest SET switching current that can be achieved and the prevents the on/off ratio from going high. This is due to the relatively high conductivity of the Sb_2Te_3 dielectric which is known to be a low-bandgap p-type semiconductor. As will be lately introduced, Sb_2Te_3 has been chosen as the dielectric in consideration of its thermal conductivity in between those of the other two control dielectrics, i.e., Bi_2Te_3 and TiTe_2 , to validate our assumption of the adversarial EC-JH in the design, as well as its desired composition, for instance no oxygen, to avoid the possibilities of forming filaments other than Te.”

To overcome this limitation and in response to your comment 4.2), **we have demonstrated a unique method to optimize our device, that is, protective electrode (PE) engineering**, in the revised manuscript. It is interesting to find that the replacement of the Pt PE with Gd PE can further push the SET switching current limit to 5 μA (**figure 1e**). **This value is comparable to that (1 μA) of the recently reported rare-earth Ru filament-based RS device.**^[2] The demonstrated low SET switching currents can be understood from the relatively low conductivity of the semiconducting Te filament compared to metallic filaments. Moreover, **it is seen that the on/off ratio increases by nearly 100 times through the use of Gd PE (figure 1e and 1f)**. Again, this is related to the semiconducting property of Te. To be specific, it is known that Te is generally p-type and therefore the use of lower WF Gd PE (2.9 eV) compared to Pt (5.9 eV) increases the Schottky barrier height at the PE/Te interface and consequently the contact resistance. As far as we know, this is the first report on the PE electrode effect on RS device performance and the exploitation of such effect.

We have added these new results and further discussions in the revised manuscript. On **page 7**, the text revision reads “To overcome this limitation, we demonstrate a unique method to reduce

the OFF state conductivity of our device, i.e., protective electrode (PE) engineering. Before we proceed, we want to point out that our TST device is sandwiched between and protected by a pair of Pt electrodes (Pt/Te/ Sb₂Te₃/Te/Pt, or PTSTP). We find that the replacement of the Pt PEs with Gd PEs (GTSTG) can increase the on/off ratio by nearly 100 times, as seen from figure 1e. This is related to the semiconducting property of Te. To be specific, it is known that Te is generally p-type and therefore the use of lower-work function Gd PE (2.9 eV) compared to Pt (5.9 eV) increases the Schottky barrier height at the PE/Te interface and consequently the contact resistance. Consistent with our expectation, this also pushes the SET switching current limit further down to 5 μ A. This value is comparable to that (1 μ A) of the recently reported rare-earth Ru filament-based RS device. In its V-RS mode, GTSTG device also has lower OFF-state conductivity compared to that of PTSTP device, as shown in figure 1f, due to the high contact resistance at the Gd/Te interface. As far as we know, this is the first report on the PE electrode effect on RS device performance and the exploitation of such effect.”

As for your comment 4.3), we apologize for a mistake left in our characterizations of the device in its NV-RS operating mode. In the revised manuscript, current compliance is only applied to the SET operations but not to the RESET operations (**figure 1c**), which is a test standard. This artifact is ruled out as a possible contribution to the NV-V transition.

As for your comment 4.4), **we would like to point out that the deduction of the current density value was based on the 2x2 μm^2 electrode size of our device.** Compared to the state-of-the-art Ag-based ECM selector device (100 μ A)^[14] with larger electrode size (5x5 μm^2), the drive current itself of our device (several mA) can still be about more than ten times larger (so at least several times larger in current density). There are indeed some reported current density values in ECM-type selectors larger than ours,^[15,16] up to 1.6 MA/cm². However, these reported devices have much smaller sizes down to several hundred nm², some thousand times smaller than ours. Moreover, because the conduction mechanism is filament-type as seen from electrode size-dependent measurements (figure 2a), the current density may well increase to a much larger value if the device is further scaled down to the nanometer scale. In fact, the currently reported record high current density values have all been achieved in OTS selector devices,^[17-19] from 20 MA/cm² to 55 MA/cm². Their sizes are also much smaller than ours by some thousand times. **As a rough guess, some tens of MA/cm² current density value can also be achieved in a Te-based V-RS device scaled down to the nanometer scale.** Experimental investigation of the scalability of the Te-RS device becomes an interesting and important future research topic.

We have added further discussions in the revised manuscript. On **page 7**, the text revision reads “On the other hand, figure 1d shows that the ON current of our device in the V-RS mode can reach 2.5 mA. Compared to the state-of-the-art Ag-based EC-RS selector device (100 μ A) with larger electrode size (5x5 μm^2), the drive current of our device is a few tens of times larger. As will be lately introduced, the conduction mechanism of our device in its LRS is filament conduction, therefore the ON current is electrode size-independent. If scaled down from the present μm -scale to some tens of nm-scale, our device could in principle delivery a few MA/cm² current density, rivalling the state-of-the-art nanoscale OTS selectors.”

Overall, in addressing the reviewer’s comments we have realized that there is large design space for Te-based RS devices and plenty of room to improve their performance. Our preliminary results become a useful springboard for more follow-up studies.

Comment 5

Does the device require a forming process before set/reset? I found this never mentioned in the manuscript.

Response

Thank you very much for your comment on this important issue. Our device does not require any forming before its reversible operation. Forming is commonly known as a process through which sufficient mobile ions are generated in the dielectric for the subsequent filamentary switching. **Forming-free property of our device may be due to the already existing Te^{2-} ions in the dielectric and therefore additional forming is not required.**

We have added further discussions in the revised manuscript. On **page 12**, the text revision reads “**Before closing this section, we would like to point out that our TST device is a forming-free device. As commonly known, forming is a process through which sufficient mobile ions are generated in the dielectric for the subsequent filamentary switching. Sufficient amount of Te^{2-} ions already exists in the Sb_2Te_3 dielectric and therefore additional forming is not required.**”

Comment 6

The abbreviation “LSL” should be defined when it is used for the first time.

Response

Thank you very much for pointing out this issue. In the revised manuscript, we have defined what abbreviation LSL stands for in the main text.

On page 14, the text revision reads “**During test, a LTP-to-STP layer (LSL) before the SNN...**”

Comment 7

The references might be too much, and the numbering in the text seems to be discontinuous.

Response

Thank you very much for pointing out this issue. We have reduced the number of the cited literatures in the revised manuscript. We also apologize for the inconvenience brought about due to our mistake in numbering the references. The mistake is corrected in the revised manuscript.

Reviewer #2

General comment

The author's manuscript shows that resistive memory switching and threshold switching are at low and high current levels, respectively, due to the low thermal conductivity and low melting point of Te in the same device. This manuscript shows its originality and novelty as following. It is explained that it works (solves the current-retention (or current-volatility) dilemma in a resistive switching device), and this result is applied to LTM-to-STM. In the reference [56] presented by the author as an example of solving the current-retention dilemma, memory and threshold switching are performed at the 1 μ A and 1mA levels, respectively. In order to operate, the graphene of the interface was formed differently. On the other hand, the results of this study show that without graphene, which is difficult to directly apply to CMOS devices, It is explained that it has high applicability because it solves the dilemma by using the same Te-based device. To be accepted, I strongly recommend authors explain sufficiently following questions and comments.

Response

Thank you very much for finding our work of originality and novelty and giving us constructive comments to further improve the quality of this work. According to your suggestions, we have fabricated new devices and carried out more thorough device characterizations and optimizations.

Our responses to your specific comments one by one are shown as follows.

Comment 1

In this manuscript, like the graphene control in [56], a key factor that can clearly distinguish between memory and threshold needs to be presented.

Response

Thank you very much for your suggestion so that the idea of this manuscript can be more clearly conveyed. A key factor that can clearly distinguish between memory switching (NV-RS) and threshold switching (V-NS) is the resistance state retention property. To be specific, in memory switching the low (LRS) and high (HRS) resistance states after SET and RESET switching, respectively, can be retained even after the voltage stress has ceased; while in threshold switching the LRS cannot be retained and will return to the initial HRS in the absence of voltage stress.

These different retention properties are reflected in the $\log|I|$ -V curves. Taking our measured $\log|I|$ -V curves as examples, when our device is operated in the memory switching mode (figure 1c), the measured current under the positive voltage sweep (from 0 V) is sharply increased to the CC limit at $\sim +1$ V and the device reaches its LRS. This is the SET process. **Then, the LRS of the device is retained under the backward voltage sweep even when the voltage has ceased, resulting in counter-clockwise hysteretic $\log|I|$ -V loop.** As the backward voltage sweep continues, the voltage polarity is reversed. When the negative voltage reaches ~ -1 V, the current rapidly drops to a low value, switching the device back to its HRS. This is the RESET process.

The retention of the LRS is a prerequisite for the occurrence of the RESET switching. The HRS of the device is then retained under the voltage sweep back to zero again, resulting in another counter-clockwise hysteretic $\log|I|$ -V loop.

On the other hand, when our device is operated in the threshold switching mode (figure 1d), the measured current under the positive voltage sweep (from 0 V) is sharply increased to the CC limit at $\sim +1$ V and the device reaches its LRS. This is similar to the SET process in the memory switching, but is commonly referred to as the threshold switching (TS) process. **Unlike memory switching, during the backward sweep of the voltage the LRS of the device can only be retained when the voltage is larger than $\sim +0.5$ V below which the device returns to its HRS.** The resulting hysteretic $\log|I|$ -V loop is also counter-clockwise. As the backward voltage sweep continues, the voltage polarity is reversed. **When the negative voltage reaches ~ -1 V, the current increases sharply, switching the device to its LRS again. This looks like a mirror process of the TS process under the positive voltage polarity.** The LRS of the device can only be retained when the absolute value of the voltage is larger than ~ 0.5 V below which the device returns to its HRS. The resulting hysteretic $\log|I|$ -V loop is clockwise and becomes a mirror loop of the counter-clockwise one under the positive voltage polarity.

In the revised manuscript, we have explained how to distinguish between memory switching and threshold switching, and have added further discussions. On **page 5**, the text revision reads “Then, the device maintains its LRS under the backward voltage sweep even after the voltage has ceased (arrow 2), resulting in counter-clockwise hysteretic $\log|I|$ -V loop. As the backward voltage sweep continues, the voltage polarity is reversed. When the negative voltage reaches ~ -1 V, the current rapidly drops to a low value, switching the device back to its high-resistance state (HRS). This is defined as the RESET process (arrow 3). It should be pointed out that the retention of the LRS is a prerequisite for the occurrence of the RESET switching.” On **page 7**, the text revisions read “Unlike NV-RS, the LRS obtained in V-RS can only be retained when a sufficiently large hold voltage is applied.” and “This looks like a mirror process of the TS process under the positive voltage polarity. Similarly, this LRS is not stable either and will return to the HRS as the voltage is swept back to a less negative value (arrow 4). The resulting hysteretic $\log|I|$ -V loop is clockwise and becomes a mirror loop of the counter-clockwise one under the positive voltage polarity.”

Comment 2

The DC memory operation window in Figure 1 is narrow. It does not seem a stable memory operation, and seems to be a temporary memory effect incidental to DC operation.

Response

Thank you very much for this useful comment. To study the retention property of the LRS obtained under the switching compliance current (CC) in the sub-mA range, we have chosen a device in its LRS obtained under 500 μ A CC as a representative test sample. **We have placed the device in ambient environment for 30 days before applying voltage with the polarity opposite to that of the voltage used to switch the device to its LRS.** It is seen from **figure S2a** that the device can still be switched back to its HRS as normal. This clearly indicates that the obtained LRS is stable and the device is indeed operated in the NV-RS mode.

In addition to this quasi-DC delayed RESET test, we have also conducted DC stress test of the LRS of the device for 10^4 seconds under constantly applied 0.1 V read voltage. No obvious degradation of the LRS has been observed (**figure S2b**), verifying that the memory effect is stable.

We have added the results of these measurements in the supplementary information of the revised manuscript.

Comment 3

Even in the explanation of the operation principle in Fig.3, the memory operation is not explicitly explained.

Response

Thank you very much for pointing out this issue. We apologize for not providing sufficient explanation of the mechanism of memory switching (NV-RS). In the revised manuscript, **figure 3** (updated) sketches the principles of operations of the Te filament-based and Ag filament-based RS devices.

For the Te filament-based device in its initial HRS (**figure 3a1**), the NV-RS process occurs if a sufficiently negative voltage is applied to the top Te active electrode (**figure 3a2**). The switching process involves the following steps:

- (i) cathodic dissolution of Te according to the reaction $\text{Te} + 2\text{e}^- \rightarrow \text{Te}^{2-}$;
- (ii) drift of Te^{2-} anions across the dielectric thin film under the action of the high electric field;
- (iii) oxidation and electro-deposition of Te on the surface of the counter electrode according to the reaction $\text{Te}^{2-} \rightarrow \text{Te} + 2\text{e}^-$.

The electro-deposition process (iii) leads to the growth of a Te filament in the direction toward the top Te electrode. After the Te filament has grown sufficiently long to make a contact to the top Te electrode, the cell is switched to the LRS (**figure 3a3**). The cell retains the LRS unless a sufficiently large voltage of opposite polarity is applied and the electrochemical dissolution of the Te filament RESETs the cell to its initial HRS.

To ensure that the Te filament-based device can be operated in the NV-RS mode, a low enough compliance current must be set so that the current passing through the growing Te filament will not be too high to fuse it instead. However, if high current is allowed to pass through (**figure 3b3**), the accumulated Joule heat could be so much that it counteracts the electrochemical (EC) effect and eventually fuses the just-grown Te filament with low melting temperature (**figure 3b4**). In this case, the operating mode of the device is transitioned to the V-RS one.

For conventional Ag filament-based device, the basic steps involved in the NV-RS process are similar to those for the Te filament-based device except that opposite voltage polarity is required for anodic dissolution of Ag to Ag^+ cations. Because Ag has much higher melting temperature than Te, high current is helpful (as long as hard breakdown does not occur) for the formation of filament because the lateral diffusion of the Ag particles and therefore the thickening of the

filament is facilitated at the elevated temperature by the JH effect (**figure 3c3**). On the contrary, if the current is low, the filament formed can be weak and unstable. Therefore, when the voltage ceases, the filament may be spontaneously ruptured, leading to V-RS (**figure 3d4**).

We have added these further explanations in the revised manuscript. On page 10, the text revision reads “For the Te filament-based device in its initial HRS (figure 3a1), the NV-RS process occurs if a sufficiently negative voltage is applied to the top Te active electrode (figure 3a2). The switching process involves the following steps:

- (i) cathodic dissolution of Te according to the reaction $\text{Te} + 2\text{e}^- \rightarrow \text{Te}^{2-}$;
- (ii) drift of Te^{2-} anions across the dielectric thin film under the action of the high electric field;
- (iii) oxidation and electro-deposition of Te on the surface of the counter electrode according to the reaction $\text{Te}^{2-} \rightarrow \text{Te} + 2\text{e}^-$.

The chemical state of Te ions involved in the electrochemical reactions is believed to be Te^{2-} which is the formal reduction state of Te. The ability to be reduced distinguishes Te from other common electrochemically active metals, such as Ag and Cu. A key evidence of the occurrence of the electrochemical reduction has been provided in a previous work where the SET switching was observed to occur only when negative voltage bias was applied to the Te electrode in the asymmetric Te/dielectric/Pt device.

The electro-deposition process (iii) leads to the growth of a Te filament in the direction toward the top Te electrode. After the Te filament has grown sufficiently long to make a contact to the top Te electrode, the cell is switched to the LRS (figure 3a3). The cell retains the LRS unless a sufficiently large voltage of opposite polarity is applied and the electrochemical dissolution of the Te filament RESETs the cell to its initial HRS.

To ensure that the Te filament-based device can be operated in the NV-RS mode, a low enough compliance current must be set so that the current passing through the growing Te filament will not be too high to fuse it instead. However, if high current is allowed to pass through (figure 3b3), the accumulated Jh becomes sufficient to fuse the just-grown Te filament with low melting temperature (figure 3b4). In this case, the operating mode of the device is transitioned to the V-RS one.

For conventional Ag filament-based device, the basic steps involved in the NV-RS process are similar to those for the Te filament-based device except that opposite voltage polarity is required for anodic dissolution of Ag to Ag^+ cations. Because Ag has much higher melting temperature than Te, high current is helpful (as long as hard breakdown does not occur) for the formation of filament because the lateral diffusion of the Ag particles and therefore the thickening of the filament is facilitated at the elevated temperature by the JH effect (figure 3c3). On the contrary, if the current is low, the filament formed can be weak and unstable. Therefore, when the voltage ceases, the filament may be spontaneously ruptured, leading to V-RS (figure 3d4).”

Comment 4

In the LTM-to-STM application, threshold switching is not focused on the application by solving the memory-threshold dilemma. Only the applicability used as a high frequency noise filter is presented. I recommend explain more detail.

Response

Thank you very much for your comment on this issue. In the original manuscript, the Te filament-based RS device solution to the memory-threshold (current-volatility or current-retention) dilemma was discussed on the basis of two separate device operation modes controlled by compliance currents (CCs). To be specific, NV-RS behavior was observed under low CCs (figure 1c and 1e), whereas V-RS behavior was observed under high CCs (figure 1d and 1f). These were in contrast to the phenomena of NV-RS under high CCs and V-RS under low CCs (memory-threshold dilemma) commonly observed in conventional ECM devices.

As has been explained in the manuscript, the memory-threshold dilemma is solved in our Te filament-based device because the low melting temperature of Te transforms the synergetic relationship between the electrochemical (EC) and Joule heating (JH) effects into adversarial one under high current. A direct result of the adversarial EC-JH effects is the unusual LTP-to-STP transition observed in the pulse train measurements under high pulse frequencies (figure 5a and 5b). To be specific, if the pulse frequency is low, the conductance of the device will gradually increase with the number of pulses, similar to the LTP phenomenon; while if the pulse frequency is high, the conductance of the device increases at the beginning but sharply decreases after a certain number of pulses due to the fusing of the just-formed Te filament by the strongly accumulated Joule heat that cannot be dissipated within such short pulse intervals. It has been based on this pulse frequency-dependent phenomenon that we proposed the high-frequency noise filter (low-pass filter) application (figure 5c).

Your comment is very helpful in that it has inspired us to further investigate the possibility to integrate two identical Te filament-based RS devices into a tandem 1S1R structure, providing direct demonstration of the dual advantages of the device as either the selector or memory. The possibility to use a single set of materials for 1S1R application may simplify the design and save otherwise substantial materials optimization effort on two types of devices with distinct operating requirements.

We have fabricated the Te/Sb₂Te₃/Te/Sb₂Te₃/Te (TSTST) structure that is composed of two identical (in the sense that the fabrication conditions and the physical dimensions are the same) Te/Sb₂Te₃/Te (TST) cells in tandem. Quasi-DC voltage sweep measurement has been performed on this TSTST device. As seen from **figure 4a**, the current undergoes obvious increase twice during the forward voltage sweep, as labeled by two arrows, respectively. After the second current jump, the current reaches its maximum set by the 1.5 mA CC. These can be understood as the typical 1S1R phenomena. To be specific, even though the two TST cells in tandem have intentionally been fabricated to be the same, in practice it is simply not possible for them to be precisely the same. This inherent device-to-device variation results in slightly faster switching-on of one device which then becomes the selector device as in the usual 1S1R structure and the delayed one becomes the memory device naturally, being responsible for the first and second current jump, respectively.

A main functional difference between our TSTST structure and the usual 1S1R structure is that in the former structure the selector or memory device is probabilistically determined but in the latter structure they are deterministically designated. Therefore, we would like to call the selector and memory devices in our TSTST structure the lucky selector (l-selector) and lucky memory (l-memory).

During the backward voltage sweep, a sharp decrease of the current is observed before the voltage has become zero. This is another typical 1S1R phenomenon that indicates the turning-off of the selector device. Admittedly, we are currently not able to verify experimentally which device is turned off, it is very likely that it is the l-selector being turned off because it has been switched on earlier and Joule heat accumulation in it is more pronounced.

After that, the $\log|I|$ -V curve retains the characteristic of high resistance (selector-off) till the sweeping voltage changes polarity. When the voltage becomes sufficiently negative, abrupt increase of current occurs once again. This is also a 1S1R phenomenon behind which is the switching-on of the bidirectional selector. If we further increase the negative voltage and then sweep it back, we notice a counter-clockwise hysteresis. This suggests that the l-memory has been reset to its HRS. Further decrease of the voltage gives rise to sharp current decrease, indicating that the l-selector has also been turned off.

1S1R behavior has been demonstrated in both Pt/TSTST/Pt (**figure 4a**) and Gd/TSTST/Gd (**figure 4b**) structures, with the latter one showing larger on/off ratio as we expected (see our responses to reviewer 1's comment #4 and your comment #6). **As far as we know, this is the first demonstration of the 1S1R behavior in tandem structure composed of two identical cells, indicating the potential of Te-based device as a universal array building block.**

Of course, from reliability consideration, practical applications may prefer deterministic selector and memory to lucky ones. In those cases, Te-based RS device can still be a promising candidate because determinacy can be engineered into the tandem structure by intentionally making the two component cells different, such as the use of different dielectric materials (figure 6) or simply different film thicknesses without changing the materials, which may still be far easier than the optimization of other conventional 1S1R devices, such as OTS: RRAM.

We have added these new results and substantial discussions in the revised manuscript. On **page 12**, the text revision reads “Because the Te-based RS device has the desired selector and memory properties which can be selectively used in the respective operating modes, a natural question arises here is whether Te-based RS device has the potential to be a universal RS cross-point array building block. To investigate, we fabricate the Te/Sb₂Te₃/Te/Sb₂Te₃/Te (TSTST) structure that is composed of two identical (in the sense that the fabrication conditions and the physical dimensions are the same) TST cells in tandem. Quasi-DC voltage sweep (from 0 V) measurements are performed on this TSTST device. As seen from figure 4a, the current undergoes obvious increase twice during the forward voltage sweep (arrow 1 and 2). After the second current jump, the current reaches its maximum set by the 1.5 mA CC. These can be understood as the typical 1S1R phenomena. To be specific, even though the two TST cells in tandem have intentionally been fabricated to be the same, in practice it is simply not possible for them to be precisely the same. This inherent device-to-device variation results in slightly faster switching-on of one device which then becomes the selector device as in the usual 1S1R structure and the delayed one becomes the memory device naturally, being responsible for the first and second current jump, respectively.

A main functional difference between our TSTST structure and the usual 1S1R structure is that in the former structure the selector or memory device is probabilistically determined but in the latter structure they are deterministically designated. Therefore, we would like to call the selector and memory devices in our TSTST structure the lucky selector (l-selector) and lucky memory (l-memory).

During the backward voltage sweep, a sharp decrease of the current is observed before the voltage has become zero. This is another typical 1S1R phenomenon that indicates the turning-off of the selector device. Admittedly, we are currently not able to verify experimentally which device is turned off, it is very likely that it is the l-selector being turned off because it has been switched on earlier and J_h accumulation in it is more pronounced.

After that, the $\log|I|$ - V curve retains the characteristic of high resistance (selector-off) till the sweeping voltage changes polarity. When the voltage becomes sufficiently negative, abrupt increase of current occurs once again. This is also a 1S1R phenomenon behind which is the switching-on of the bidirectional selector. If we further increase the negative voltage and then sweep it back, we notice a counter-clockwise hysteresis. This suggests that the memory has been reset to its HRS. Further decrease of the voltage gives rise to sharp current decrease, indicating that the selector has also been turned off. Supplementary figure S7 sketches the principles of 1S1R operations of TSTST device.

Similar 1S1R characteristics are also observed in tandem device with the Gd PEs. The on/off ratio is larger as expected, as seen from figure 4b. As far as we know, this is the first demonstration of the 1S1R behavior in tandem structure composed of two identical cells. It is worth noting that, from reliability consideration, practical applications may prefer deterministic selector and memory to lucky ones. In those cases, Te-based RS device can still be a promising candidate because determinacy can be engineered into the tandem structure by intentionally making the two component cells different, such as the use of different dielectric materials (as will lately been seen in figure 6) or simply different film thicknesses without changing the materials, which may still be far easier than the optimization of other conventional 1S1R devices, such as OTS: RRAM. The possibility to use a single set of materials for 1S1R application may simplify the design and save otherwise substantial materials optimization effort on two types of devices with distinct operating requirements.”

Comment 5

Since the temperature distribution in Fig. 6 explains the contention between memory-threshold operations, it can be explained that Sb₂Te₃ is in the transition period of the two switching modes. However, it is understood that it is difficult to operate the two operations independently. (Related to comment #1)

Response

Thank you very much for your comment. We try to address it from two aspects.

As a standalone device, our Te/Sb₂Te₃/Te (TST) device can be operated under two different modes selectively by setting the compliance current (CC): NV-RS (memory) mode under low CC and V-RS (selector) mode under high CC. A key factor that can clearly distinguish between

NV-RS and V-RS has been presented in addressing comment 1. As the reviewer has pointed out, there is contention between these two operation modes as shown in figure 6c. To be specific, such contention becomes intense as the CC increases to heat up the Te filament to the vicinity of its melting temperature. This threshold CC will increase with the thermal conductivity of the dielectric where the Te filament is embedded.

However, we do not consider the contention between these two modes to be problematic in real selector and memory applications. This is because memory devices are desired to be operated under low CC while selector devices are preferred to be operated under high CC. These properties are exactly what Te filament-based device can offer. **As shown in figure 1, our device in its NV-RS mode can be operated under quite low CC about several μA and in its V-RS mode under quite high CC about several mA. The 10^3 order of magnitude CC difference between these two operation modes reduces the chances of their contention and ensures reliability. On the contrary, the contention would be problematic for a conventional metal filament-based RS device because the more we push the limits for metallic filament-based selectors and memories the closer are they to the NV-RS-to-V-RS transition point where undesired transition between NV-RS and V-RS may occur.**

We have added these discussions in the revised manuscript. On **page 17**, the text revision reads “In practical memory and selector applications, the Te filament-based RS devices can be more reliably operated near their respective performance limits, i.e., lowest and highest achievable operating currents, than the conventional metallic filament-based RS devices in the sense that the undesired transition between NV-RS and V-RS is less likely to occur in the former devices. This is because the more we push the limits for Te filament-based selectors and memories the more distant are they away from the NV-RS-to-V-RS transition point. On the contrary, due to the current-volatility dilemma, the limits for metallic filament-based selectors and memories are near the transition point where undesired transition between NV-RS and V-RS may occur.”

From another perspective, considering integrated and fully functioned 1S1R structure, we have demonstrated 1S1R behavior in tandem structure Te/Sb₂Te₃/Te/Sb₂Te₃/Te (TSTST) composed of two identical TST cells (see our responses to your comment #4 and figure 4). **This is a strong indication that the dual modes that can only be selectively activated in a standalone device can also function simultaneously by the integration of two devices, even if the devices are identical in terms of constituent materials and physical dimensions.** Further optimizations of the Te-based 1S1R device can be performed by using dielectric materials with different thermal conductivities for the selector and memory components (figure 6) or simply using different film thicknesses without changing the materials, which may be far easier than the optimization of other conventional 1S1R devices, such as OTS: RRAM. **As far as we know, this is the first demonstration of the 1S1R behavior in tandem structure composed of two identical cells, indicating the potential of Te-based device as a universal array building block.** The possibility to use a single set of materials for 1S1R application may simplify the design and save otherwise substantial materials optimization effort on two types of devices with distinct operating requirements.

We have added these discussions in the revised manuscript. On **page 13**, the text revision reads “It is worth noting that, from reliability consideration, practical applications may prefer deterministic selector and memory to lucky ones. In those cases, Te-based RS device can still be

a promising candidate because determinacy can be engineered into the tandem structure by intentionally making the two component cells different, such as the use of different dielectric materials (as will lately been seen in figure 6) or simply different film thicknesses without changing the materials, which may still be far easier than the optimization of other conventional 1S1R devices, such as OTS: RRAM.”

Comment 6

Memory On/Off Ratio is too small to support memory operation, which is critical for real array operation.

Response

Thank you very much for raising this important comment which has inspired us to further explore the design space of our Te device. **First, we would like to explain why the memory on/off ratio is small in our (Pt)Te/Sb₂Te₃/Te(Pt) device. In this work, Sb₂Te₃ has been chosen as the dielectric. It is known to be a low-bandgap p-type semiconductor. Therefore, its conductivity is relatively high. We have chosen Sb₂Te₃ in consideration of its thermal conductivity in between those of other two control dielectrics, i.e., Bi₂Te₃ and TiTe₂, to validate our assumption of the adversarial EC-JH in the design, as well as its desired composition, for instance no oxygen, to avoid the possibilities of forming filaments other than Te.**

To overcome this limitation, we have demonstrated a unique method to optimize our device, that is, protective electrode (PE) engineering, in the revised manuscript. It is seen that the on/off ratio increases by nearly 100 times through the use of Gd PE (figure 1e and 1f). This is related to the semiconducting property of Te. To be specific, it is known that Te is generally p-type and therefore the use of lower WF Gd PE (2.9 eV) compared to Pt (5.9 eV) increases the Schottky barrier height at the PE/Te interface and consequently the contact resistance. **As far as we know, this is the first report on the PE electrode effect on RS device performance and the exploitation of such effect.**

We have added these new results and further discussions in the revised manuscript. On page 7, the text revision reads “As can be seen from figure 1c, our proof-of-concept device has relatively high OFF current which sets a fundamental limit on the lowest SET switching current that can be achieved and the prevents the on/off ratio from going high. This is due to the relatively high conductivity of the Sb₂Te₃ dielectric which is known to be a low-bandgap p-type semiconductor. As will be lately introduced, Sb₂Te₃ has been chosen as the dielectric in consideration of its thermal conductivity in between those of other two control dielectrics, i.e., Bi₂Te₃ and TiTe₂, to validate our assumption of the adversarial EC-JH in the design, as well as its desired composition, for instance no oxygen, to avoid the possibilities of forming filaments other than Te.

To overcome this limitation, we demonstrate a unique method to reduce the OFF state conductivity of our device, i.e., protective electrode (PE) engineering. Before we proceed, we want to point out that our TST device is sandwiched between and protected by a pair of Pt electrodes (Pt/Te/ Sb₂Te₃/Te/Pt, or PTSTP). We find that the replacement of the Pt PEs with Gd PEs (GTSTG) can increase the on/off ratio by nearly 100 times, as seen from figure 1e. This is related to the semiconducting property of Te. To be specific, it is known that Te is generally p-

type and therefore the use of lower-work function Gd PE (2.9 eV) compared to Pt (5.9 eV) increases the Schottky barrier height at the PE/Te interface and consequently the contact resistance. Consistent with our expectation, this also pushes the SET switching current limit further down to 5 μA . This value is comparable to that (1 μA) of the recently reported rare-earth Ru filament-based RS device. In its V-RS mode, GTSTG device also has lower OFF-state conductivity compared to that of PTSTP device, as shown in figure 1f, due to the high contact resistance at the Gd/Te interface. As far as we know, this is the first report on the PE electrode effect on RS device performance and the exploitation of such effect.

Reviewer #3

General comment

The authors proposed an interesting approach to overcome the current-volatility dilemma of electrochemical resistive switching (RS) devices by utilizing a Te filament in a Te/Sb₂Te₃/Te structure. Semiconducting Te has lower thermal conductivity and lower melting temperature than metal materials (such as Cu and Ag) usually adopted in electrochemical RS devices. Owing to the stronger thermal confinement and higher local temperature around the Te filament, the device exhibits non-volatile RS (NV-RS) with low ON current and volatile RS (V-RS) with high ON current, which fulfill the requirements for memories and selectors in cross-point arrays. The authors also demonstrate unusual stimulus frequency dependent long-term memory to short-term memory transition under voltage pulse trains and tunable switching behaviors between NV-RS and V-RS by changing the telluride matrix. The paper is well written and contains new results of reasonable significance. However, I would like to have the authors to consider the following issues and respond to them.

Response

Thank you very much for finding our work of significance and giving us constructive comments to further improve the quality of this work. According to your suggestions, we have fabricated new devices and carried out more thorough device characterizations and optimizations.

Our responses to your specific comments one by one are shown as follows.

Comment 1

Throughout the paper, the style of figures is incorrect and confusing. The alphabetical characters (a, b, c, etc.) in the bottom left-hand side corner should be moved to the upper left-side corner.

Response

Thank you very much for pointing out these issues to improve the readability of our manuscript. We have thoroughly checked these problems and made substantial improvements.

Comment 2

The elemental mapping of TEM image (Fig. 1b) is difficult to recognize where the Te/Sb₂Te₃ and Sb₂Te₃/Te interfaces are located. Why the authors show the images with different magnifications in Figs. 1a and 1b?

Response

Thank you very much for your comment to make us aware of these problems. In **figure 1b** of the revised manuscript, we have presented the result of the elemental mapping by EDS line scanning along the yellow line as denoted in figure 1a. The boundary between Sb₂Te₃/Te can now be identified clearer. We have also replaced the originally superimposed color coded elemental map, in which the boundary has been blurred, with two separate ones in **figure S1**.

We have originally shown images with different magnifications from layout consideration. We have made changes in the revised manuscript. The corresponding texts have also been changed. On page 5, the text revision reads “The results of the elemental mapping by energy-dispersive X-ray spectroscopy (EDS) line scanning along the yellow line as denoted in figure 1a are shown in figure 1b and supplementary figure S1.”

Comment 3

In NV-RS shown in Figs. 1c and 1d, the current compliance is always set to a higher value in the negative bias than in the positive bias. What is the role of the current compliance in the negative bias? Is it necessary to realize the NV-RS behavior?

Response

Thank you very much for pointing out this issue. We apologize for this mistake left in our characterizations of the device in its NV-RS operating mode. In the revised manuscript, current compliance is only applied to the SET operations but not to the RESET operations (**figure 1c**), which is a test standard. This artifact is ruled out as a possible contribution to the NV-V transition. We have updated the results in the revised manuscript.

Comment 4

In the simulations of the temperature distribution, as shown in Figs. 2d and 2e, there is no information about the filament size. The authors should describe the diameter on both side and the filament length in the text. The scale bar should also be indicated in the figures.

Response

Thank you very much for your comment to improve the scientific rigor of the presentations. In the revised manuscript, we have added these information in the revised manuscript and added scale bar to the figures (moved to the supplementary information as figure S6 for better structured manuscript). On **page 10**, the text revision reads “30 nm in height, 8 nm and 3 nm in diameter for the top and bottom surfaces, respectively”

Comment 5

Although the authors discussed the difference between Ag filament-based and Te filament-based devices in Fig. 2d and 2e, the explanation of Fig. 3 is not enough. I think the authors should discuss the difference in switching mechanism between Ag filament-based and Te filament-based RS memories in more detail using Fig. 3. Otherwise, Fig. 3 should be moved to Supplementary Information.

Response

Thank you very much for your suggestions to discuss in more details on the difference between Te-based filamentary switching and Ag-based filamentary switching.

In the revised manuscript, **figure 3** (updated) sketches the principles of operations of the Te filament-based and Ag filament-based RS devices.

For the Te filament-based device in its initial HRS (**figure 3a1**), the NV-RS process occurs if a sufficiently negative voltage is applied to the top Te active electrode (**figure 3a2**). The switching process involves the following steps:

- (i) cathodic dissolution of Te according to the reaction $\text{Te} + 2\text{e}^- \rightarrow \text{Te}^{2-}$;
- (ii) drift of Te^{2-} anions across the dielectric thin film under the action of the high electric field;
- (iii) oxidation and electro-deposition of Te on the surface of the counter electrode according to the reaction $\text{Te}^{2-} \rightarrow \text{Te} + 2\text{e}^-$.

The electro-deposition process (iii) leads to the growth of a Te filament in the direction toward the top Te electrode. After the Te filament has grown sufficiently long to make a contact to the top Te electrode, the cell is switched to the LRS (**figure 3a3**). The cell retains the LRS unless a sufficiently large voltage of opposite polarity is applied and the electrochemical dissolution of the Te filament RESETs the cell to its initial HRS.

To ensure that the Te filament-based device can be operated in the NV-RS mode, a low enough compliance current must be set so that the current passing through the growing Te filament will not be too high to fuse it instead. However, if high current is allowed to pass through (**figure 3b3**), the accumulated Joule heat could be so much that it counteracts the electrochemical (EC) effect and eventually fuses the just-grown Te filament with low melting temperature (**figure 3b4**).

For conventional Ag filament-based device, the basic steps involved in the NV-RS process are similar to those for the Te filament-based device except that opposite voltage polarity is required for anodic dissolution of Ag to Ag^+ cations. Because Ag has much higher melting temperature than Te, high current is helpful (as long as hard breakdown does not occur) for the formation of filament because the lateral diffusion of the Ag particles and therefore the thickening of the filament is facilitated at the elevated temperature by the JH effect (**figure 3c3**). On the contrary, if the current is low, the filament formed can be weak and unstable. Therefore, when the voltage ceases the filament may be spontaneously ruptured, leading to V-RS (**figure 3d4**).

We have added these further explanations in the revised manuscript. On page 10, the text revision reads “For the Te filament-based device in its initial HRS (**figure 3a1**), the NV-RS process occurs if a sufficiently negative voltage is applied to the top Te active electrode (**figure 3a2**). The switching process involves the following steps:

- (i) cathodic dissolution of Te according to the reaction $\text{Te} + 2\text{e}^- \rightarrow \text{Te}^{2-}$;
- (ii) drift of Te^{2-} anions across the dielectric thin film under the action of the high electric field;
- (iii) oxidation and electro-deposition of Te on the surface of the counter electrode according to the reaction $\text{Te}^{2-} \rightarrow \text{Te} + 2\text{e}^-$.

The chemical state of Te ions involved in the electrochemical reactions is believed to be Te^{2-} which is the formal reduction state of Te. The ability to be reduced distinguishes Te from other common electrochemically active metals, such as Ag and Cu. A key evidence of the occurrence of the electrochemical reduction has been provided in a previous work where the SET switching

was observed to occur only when negative voltage bias was applied to the Te electrode in the asymmetric Te/dielectric/Pt device.

The electro-deposition process (iii) leads to the growth of a Te filament in the direction toward the top Te electrode. After the Te filament has grown sufficiently long to make a contact to the top Te electrode, the cell is switched to the LRS (figure 3a3). The cell retains the LRS unless a sufficiently large voltage of opposite polarity is applied and the electrochemical dissolution of the Te filament RESETs the cell to its initial HRS.

To ensure that the Te filament-based device can be operated in the NV-RS mode, a low enough compliance current must be set so that the current passing through the growing Te filament will not be too high to fuse it instead. However, if high current is allowed to pass through (figure 3b3), the accumulated Jh becomes sufficient to fuse the just-grown Te filament with low melting temperature (figure 3b4). In this case, the operating mode of the device is transitioned to the V-RS one.

For conventional Ag filament-based device, the basic steps involved in the NV-RS process are similar to those for the Te filament-based device except that opposite voltage polarity is required for anodic dissolution of Ag to Ag⁺ cations. Because Ag has much higher melting temperature than Te, high current is helpful (as long as hard breakdown does not occur) for the formation of filament because the lateral diffusion of the Ag particles and therefore the thickening of the filament is facilitated at the elevated temperature by the JH effect (figure 3c3). On the contrary, if the current is low, the filament formed can be weak and unstable. Therefore, when the voltage ceases, the filament may be spontaneously ruptured, leading to V-RS (figure 3d4).”

Comment 6

On page 9, line 3: '1 μs' should be '10 μs'.

Response

Thank you very much. We have updated the texts (in the revised manuscript, this part has been moved to the supplementary information as figure S4 for better structured manuscript)

Comment 7

The endurance results under voltage pulse trains (Figs. 4c and 4d) show that the ON and OFF resistances decrease with increasing the number of voltage pulses. These behaviors imply that the degradation of the device performance takes place in the pulse train measurements even with 1 μs pulse width. How many cycles can the device repeat RS? The endurance data for more cycles had better put in Supplementary Information.

Response

Thank you very much for your suggestions so as to make this manuscript more comprehensive. Results of longer pulse train measurements have been given in **figure S4** of the revised manuscript for Pt and Gd protected devices, respectively. It is seen that while the state of low resistance (LR) is reasonably stable, the state of high resistance (HR) suffers from certain degree

of degradation with increasing number of stressing pulses. This looks more severe for the device operated in the V-RS mode, which can be understood from the more intense stress applied. Similar phenomena have also been frequently observed in conventional Ag and Cu-based ECM cells and **have generally been attributed to filament overgrowth.**^[5]

It is also interesting to find that the replacement of the Pt protective electrode (PE) with Gd PE, which has originally aimed to increase the dynamic range of the RS (in response to reviewer 1's comment #4 and reviewer 2's comment #6), enhances the endurance performance. **It is likely that the better performance of Gd protected device originates from the smaller electronegativity of Gd (1.21) compared to Pt (2.28) and the consequent stronger binding to Te. This may suppress the injection of excessive Te²⁻ into the dielectric and therefore mitigate filament overgrowth** which is generally believed to be a main culprit for the endurance issue.^[5]

We have updated the texts regarding the pulse endurance tests and have added further discussions in the revised manuscript. On **page 8**, the text revision reads “**Endurance tests have also been performed for both PTSTP and GTSTG devices in their respective NV-RS and V-RS operating modes, as shown in supplementary figure S3 and S4. Continuous quasi-DC sweeps and pulse train measurements show certain degree of on/off ratio degradation. Similar phenomena have also been frequently observed in conventional Ag and Cu-based ECM cells and have generally been attributed to filament overgrowth. In this respect, it is interesting to find that GTSTG device has better endurance performance than PTSTP device. It is likely that the better performance of Gd protected device originates from the smaller electronegativity of Gd (1.21) compared to Pt (2.28) and the consequent stronger binding to Te. This may suppress the injection of excessive Te²⁻ into the dielectric and therefore mitigate filament overgrowth.**”

Comment 8

In the simulations of the temperature distribution shown in Fig. 6d, the diameter and length of the filament should be described. Is the thermal conductivity of the Te electrode on both sides taken into account (also for the results of Figs. 2d and 2e)? The detailed conditions in the simulations may be better to described in the Experimental section or in Supplementary Information.

Response

Thank you very much for your suggestions in order to improve the scientific rigor of our manuscript. We have described the diameter and length of the simulated filament in the revised manuscript. On **page 17**, the text revision reads “**1 nm in diameter, 30 nm in length**”

The thermal conductivity of the Te electrode on both sides has been taken into consideration in all the COMSOL simulations in this work. In the revised manuscript, we have clarified this issue and provided more detailed information on the conditions used in the simulations in the experimental section.

References

- [1] Yoon, J. H., Zhang, J. M., Ren, X. C., Wang, Z. G., Wu, H. Q., Li, Z. Y., Barnell, Mark., Wu, Q., Lauhon, L. J., Xia, Q. F. & Yang, J. J. Truly Electroforming-Free and Low-Energy Memristors with Preconditioned Conductive Tunneling Paths. *Adv. Funct. Mater.* **27**, 1702010 (2017).
- [2] Yoon, J. H., Zhang, J. M., Lin, P., Upadhyay, N., Yan, P., Liu, Y. Z., Xia, Q. F. & Yang, J. J. A Low-Current and Analog Memristor with Ru as Mobile Species. *Adv. Mater.* **32**, 1904599 (2020).
- [3] Zhang, Z. Y., Wang, Y. Y., Luo, Y., He, Y. H., Ma, M. Y., Yang, R. R. & Li, H. L. Electrochemical metallization cell with anion supplying active electrode. *Sci. Rep.* **8**, 12617 (2018).
- [4] Zhao, X. L., Ma, J., Xiao, X. H., Liu, Q., Shao, L., Chen, D., Liu, S., Niu, J. B., Zhang, X. M., Wang, Y., Cao, R. R., Wang, W., Di, Z. F., Lv, H. B., Long, S. B. & Liu, M. Breaking the Current-Retention Dilemma in Cation-Based Resistive Switching Devices Utilizing Graphene with Controlled Defects. *Adv. Mater.* **30**, 1705193 (2018).
- [5] Liu, S., Lu, N. D., Zhao, X. L., Xu, H., Banerjee, W., Lv, H. B., Long, S. B., Li, Q. J., Liu, Q. & Liu, M. Eliminating Negative-SET Behavior by Suppressing Nanofilament Overgrowth in Cation-Based Memory. *Adv. Mater.* **28**, 10623-10629 (2016).
- [6] Fujii, S., Incorvia, J. A. C., Yuan, F., Qin, S. J., Hui, F., Shi, Y. Y., Chai, Y., Lanza, M. & Wong, H.-S. P. Scaling the CBRAM Switching Layer Diameter to 30 nm Improves Cycling Endurance. *IEEE Electron Device Lett.* **39**, 23-26 (2017).
- [7] Li, Y., Long, S. B., Liu, Q., Lv, H. B. & Liu, M. Resistive Switching Performance Improvement via Modulating Nanoscale Conductive Filament, Involving the Application of Two-Dimensional Layered Materials. *Nano Micro Small* **13**, 1604306 (2017).
- [8] Tao, Y., Li, X. H., Xu, H. Y., Wang, Z. Q., Ding, W. T., Liu, W. Z., Ma, J. G. & Liu, Y. C. Improved Uniformity and Endurance Through Suppression of Filament Overgrowth in Electrochemical Metallization Memory With AgInSbTe Buffer Layer. *IEEE J. Electron Devices Soc.* **6**, 714-720 (2018).
- [9] Cao, R. R., Liu, S., Liu, Q., Zhao, X. L., Wang, W., Zhang, X. M., Wu, F. C., Wu, T. T., Wang, Y., Lv, H. B., Long, S. B. & Liu, M. Improvement of Device Reliability by Introducing a BEOL-Compatible TiN Barrier Layer in CBRAM. *IEEE Electron Device Lett.* **38**, 1371-1374 (2017).
- [10] Kaeriyama, S., Sakamoto, T., Sunamura, H., Mizuno, M., Kawaura, H., Hasegawa, T., Terabe, K., Nakayama, T. & Aono, M. A nonvolatile programmable solid-electrolyte nanometer switch. *IEEE Journal of Solid-State Circuits* **40**, 168-176 (2005).
- [11] Wu, T. T., Banerjee, W., Cao, J. C., Ji, Z. Y., Li, L. & Liu, M. Improvement of durability and switching speed by incorporating nanocrystals in the HfO_x based resistive random access memory devices. *Appl. Phys. Lett.* **113**, 023105 (2018).
- [12] Kwon, D-H., Kim, K. M., Jang, J. H., Jeon, J. M., Lee, M. H., Kim, G. H., Li, X-S., Park, G-S., Lee, B., Han, S., Kim, M. & Hwang, C. S. Atomic structure of conducting nanofilaments in TiO₂ resistive switching memory. *Nat. Nanotechnol.* **5**, 148-153 (2010).
- [13] Yang, Y. C., Gao, P., Li, L. Z., Pan, X. Q., Tappertzhofen, S., Choi, S. H., Waser, R., Valov, I. & Lu, W. D. Electrochemical dynamics of nanoscale metallic inclusions in dielectrics. *Nat. Commun.* **5**, 4232 (2014).
- [14] Midya, R., Wang, Z. R., Zhang, J. M., Savel'ev, S. E., Li, C., Rao, M. Y., Jang, M. H., Joshi, S., Jiang, H., Lin, P., Norris, K., Ge, N., Wu, Q., Barnell, M., Xin, L. H., Williams, R. S., Xia, Q. F. & Yang, J. J. Anatomy of Ag/Hafnia-Based Selectors with 10¹⁰ Nonlinearity. *Adv. Mater.* **29**, 1604457 (2017).
- [15] Han, U. B., Lee, D. & Lee, J.S. Reliable current changes with selectivity ratio above 10⁹ observed in lightly doped zinc oxide films. *NPG Asia Mater.* **9**, e351 (2017).
- [16] Bricalli, A., Ambrosi, E., Laudato, M., Maestro, M., Rodriguez, R. & Ielmini, D. SiO_x-based resistive switching memory (RRAM) for crossbar storage/select elements with high on/off ratio. In *2016 IEEE Int. Electron Devices Meeting (IEDM)* on 4.3.1-4.3.4.
- [17] Govoreanu, B., Donadio, G. L., Opsomer, K., Devulder, W., Afanas'ev, V. V., Witters, T., Clima, S., Avsarala, N. S., Redolfi, A., Kundu, S., Richard, O., Tsvetanova, D., Pourtois, G., Detavemier, C., Goux, L. & Kar, G. S. Thermally stable integrated Se-based OTS selectors with >20 MA/cm² current

drive, $>3 \cdot 10^3$ half-bias nonlinearity, tunable threshold voltage and excellent endurance. in *2017 Symp. VLSI Technology (VLSIT)* on T92-T93.

[18] Jia, S. J., Li, H. L., Gotoh, T., Longeaud, C., Zhang, B., Lyu, J., Lv, S. L., Zhu, M., Song, Z. T., Liu, Q., Robertson, J. & Liu, M. Ultrahigh drive current and large selectivity in GeS selector. *Nat. Commun.* **11**,4636 (2020).

[19] Yoo, J., Lee, D., Park, J., Song, J. & Hwang, H. Steep Slope Field-Effect Transistors With B-Te-Based Ovonic Threshold Switch Device. *IEEE J. Electron Devices Soc.* **6**, 821-824 (2018).

Reviewers' Comments:

Reviewer #1:

Remarks to the Author:

The authors have responded to previous comments and made some improvements in the revised manuscript. However, some issues are still not addressed properly:

1. I found the chemical state of Te during electrochemical reactions is not clearly discussed. The chemical state of Te cannot be fixed as Te^{2-} if they are electrochemically active, but goes through repetitive oxidation/reduction processes, where the chemical state of Te changes. The authors should discuss the evolution of state during the resistive switching process instead.
2. The switching current of 25 μA after optimization is NOT among the lowest for electrochemical metallization RS devices. For example, APPLIED PHYSICS LETTERS 92, 122910, 2008 shows switching current of 10 pA, which is 6 orders of magnitude lower. There are a lot more studies reporting low switching currents.
3. The DC switching cycles of 200 after optimization is still too low. The authors cited Ref. 4, but finding a similarly low endurance value in the literature does not mean this is what is required in applications. The authors performed pulse measurements to show 8k endurance cycles, but as the authors stated the degradation effect is quite obvious.
4. The authors performed TEM characterization but actually no meaningful data were shown. The switching mechanism is still largely based on assumptions.
5. The authors didn't address the question regarding the different off resistance values directly. It is not about the current compliance. Instead, the nonvolatile (Fig. 1c,e) mode generally shows lower OFF-state resistance than the volatile (Fig. 1d,f) mode. Will this contribute to the different switching behaviors as well?

Reviewer #2:

None

Reviewer #3:

Remarks to the Author:

The manuscript has been improved along my comments and suggestions. The authors have also added new data to the revised manuscript, in response to the comments of other reviewers, which increases the quality of the paper. Thus, I recommend publication in Nature Communications. Please revise the following minor mistakes.

- 1) On page 4, line 122: '452 ' should be '452°C'.
- 2) The abbreviation of the Joule heating (JH) is denoted to 'Jh' in many places, such as line 277, 279, 285 of page 10, line 314 of page 11, line 363 of page 12, line 394 of page 13, line 401, 402 of page 14, etc.

Point-by-point response to reviewers' comments

Reviewer #1

General comment

The authors have responded to previous comments and made some improvements in the revised manuscript. However, some issues are still not addressed properly.

Response

Thank you very much again for spending your valuable time on reviewing our manuscript and providing professional suggestions to help further improve the quality of the manuscript.

According to your comments on our previously revised manuscript R1, we here have

- 1) performed temperature dependent electrical conduction measurements to provide additional evidence on the semiconducting property of the filament material,
- 2) fabricated 60 nm device by e-beam lithography to achieve switching current as low as 50 pA in the NV-RS mode,
- 3) investigated the scaling effects on the endurance performance of the device and found that the endurance could be improved by down-scaling the device to nm scale.

Our responses to your specific comments one by one are shown as follows.

Comment 1

I found the chemical state of Te during electrochemical reactions is not clearly discussed. The chemical state of Te cannot be fixed as Te^{2-} if they are electrochemically active, but goes through repetitive oxidation/reduction processes, where the chemical state of Te changes. The authors should discuss the evolution of state during the resistive switching process instead.

Response

We would like to thank the reviewer for pointing out this issue that is important for the readers to better understand the resistive switching mechanism of our device. **In the previously revised manuscript R1, we have already added discussions** on the chemical state evolution of Te during the SET (threshold switching is similar) switching which involves successive Te reduction and Te^{2-} oxidation processes (see page 10 of R1 or page 11 of R2).

In this revised manuscript R2, we have added further discussions on the chemical state evolution of Te during the RESET switching which also involves successive Te reduction and Te^{2-} oxidation processes, and presented the mechanisms schematically in the supplementary information (**supplementary figure S10**). **Figure 3 has also been updated by explicitly showing the chemical state evolution of Te during the dynamic processes.** On page 11, the text revision reads “During RESET, the Te filament is dissolved by reversing the electrochemical processes:

(i) dissolution of the Te filament at its end near the bottom electrode according to the reaction $Te+2e^{-}\rightarrow Te^{2-}$;

- (ii) drift of Te^{2-} anions across the dielectric thin film under the action of the high electric field;
- (iii) oxidation of Te^{2-} on the surface of the top electrode according to the reaction $\text{Te}^{2-} \rightarrow \text{Te} + 2\text{e}^-$.

Note that in the initial phase of RESET there has already been current through the Te filament. The generated Joule heat assists the out-diffusion and drift of the dissolved Te^{2-} anions. The rupture of the Te filament at one end or at its weakest point (process i) results in sudden increase of the device resistance and consequently sharp decrease of the current. This terminates the rupture process and leaves the main body of the filament intact. Therefore, the successive SET switching may only be a matter of restoring the tiny ruptured point instead of reconstructing the whole filament.”

Comment 2

The switching current of 25 μA after optimization is NOT among the lowest for electrochemical metallization RS devices. For example, APPLIED PHYSICS LETTERS 92, 122910, 2008 shows switching current of 10 pA, which is 6 orders of magnitude lower. There are a lot more studies reporting low switching currents.

Response

Thank you very much for pointing out an inaccurate statement in our previous response letter. In reaching the conclusion that our switching current is among the lowest, we have had the comparison group of ECM devices with the sizes of the order of μm^2 , comparable to our $2 \times 2 \mu\text{m}^2$. Under this condition, the difference in the amplitude of the switching current is mainly determined by the conductivity of the filament. The low current of our device can be understood from the relatively low conductivity of the semiconducting Te filament compared to metallic filaments.

In addition to the conductivity of the filament material, the OFF state conductivity of the device sets a fundamental limit on the lowest SET switching current that can be achieved. Usually, the OFF state conductivity is monotonically dependent on the size of the device. Therefore, shrinking the size of the device can in principle be a route toward even lower switching current. In fact, the device area in *Appl. Phys. Lett.* 92, 122910 (2008) was about 500 times smaller than ours, with the size of $86 \times 86 \text{ nm}^2$.

In this revised manuscript R2, we have fabricated device of the diameter of 60 nm by e-beam lithography and performed electrical measurements. **SET switching current as low as 50 pA can be achieved** (in the NV-RS mode), as can be seen in **supplementary figure S3**. In addition to these new results presented in the main text, we have also specified the sizes of other reported ECM devices in the comparison group, and added discussions on the reduction of the switching current. On **page 8**, the text revision reads “Further decrease of the OFF state conductivity can be achieved by down-scaling the device to nm scale at which the SET switching current could be even lower. T-shape TiN/Te/Sb₂Te₃/Te/TiN (T’TSTT’) device of the diameter of 60 nm is fabricated by e-beam lithography. Ultralow SET switching current of 50 pA can be achieved, as shown in supplementary figure S3. It is anticipated that by replacing Sb₂Te₃ with other wide-bandgap insulator, such as SiO₂ and Ta₂O₅, the OFF state conductivity and consequently the SET switching current can be further reduced”.

In addition to size shrinkage, materials optimizations are also promising toward even lower SET switching current. These include the replacement of Sb_2Te_3 with other wide-bandgap insulator, such as SiO_2 and Ta_2O_5 , and the use of low-work function protective electrode, such as Gd, as has been demonstrated in our previously revised manuscript R1 (figure 1e). The rationale behind the latter approach is Schottky barrier height modulation, considering the semiconducting property of Te.

Comment 3

The DC switching cycles of 200 after optimization is still too low. The authors cited Ref. 4, but finding a similarly low endurance value in the literature does not mean this is what is required in applications. The authors performed pulse measurements to show 8k endurance cycles, but as the authors stated the degradation effect is quite obvious.

Response

Thank you very much for your comment. We cannot agree more that endurance is one of the important reliability performance metrics for electronic devices. In the previously revised manuscript R1, we have found that the replacement of Pt protective electrode with less electronegative Gd can enhance the endurance performance to a certain extent. This has been understood from the suppression of the injection of excessive Te^{2-} into the dielectric due to stronger Gd-Te binding and therefore mitigate filament overgrowth.

In this revised manuscript R2, we have further investigated the scalability of the device and the scaling effect on the endurance performance of the device. To this end, we have fabricated T-shape TiN/Te/Sb₂Te₃/Te/TiN device of the diameter of 150 nm and conducted endurance measurements on this device. It is seen that the endurance of the down-scaled device enhances compared to the μm -scale device (**supplementary figure S6**). This can be understood from not only the larger dynamic range that is more resistance degradation tolerable, but also the more spatially confined Te^{2-} injection into the dielectric.

We have presented these new results and added further discussions in this revised manuscript R2. On **page 8**, the text revision reads “In addition to this nonconventional PE engineering method, there are at least two other possible solutions to optimize the endurance performance of the Te-based RS device, i.e., scaling down the device and optimizing the dielectric layer. Currently, our device is $2 \times 2 \mu\text{m}^2$. Scaling it further down will limit Te supply and confine Te^{2-} injection into the dielectric layer which may improve its endurance. In addition to spatial confinement of Te^{2-} injection, optimizing the dielectric layer by using less Te-dissolvable dielectric or inserting appropriate Te^{2-} ion buffer layer is also considered as a viable solution. Here, we investigate the effects of scaling. Supplementary figure S6 shows the endurance performance of a 150 nm T’TSTT’ device under pulse train stimuli. As expected, the endurance of the down-scaled device enhances compared to the μm -scale device, not only because of the larger dynamic range that is more resistance degradation tolerable but also because of the more spatially confined Te^{2-} injection into the dielectric”.

Comment 4

The authors performed TEM characterization but actually no meaningful data were shown. The switching mechanism is still largely based on assumptions.

Response

Thank you for your comment. On this point, however, we do not think that the proposed switching mechanism is largely based on assumptions. **Instead, it has been supported by a number of experimental evidences:**

1) Area dependent conductivity measurements, which are standardized and widely used means to verify one-dimensional filamentary conduction, have already provided strong evidence of the filamentary nature of the RS (figure 2a);

2) TEM observations have provided direct evidence of the existence of a Te filament after SET switching (figure 2b). In trying to rule out the possibility of pre-existing filament(s) in the as-fabricated device, we have carried out extensive (though not exhaustive) examinations of the cross-section areas of the as-fabricated device and found no filament (supplementary figure S8).

3) In this revised manuscript R2, temperature dependent electrical conduction measurements have been conducted, as shown in supplementary figure S7. The conductivity of our Te filament-based device after SET switching increases with temperature, indicating the semiconducting property of the local filament (Te is a semiconductor). **In comparison, our fabricated Pt/Ag/Sb₂Te₃/Ag/Pt ECM device shows negative correlation between its SET-state conductivity and temperature, in consistence with the metallic property of Ag filament.** We have added text describing these results. On **page 9**, the text revision reads “Temperature dependent conductivity measurements are also conducted on two control devices in their respective SET states, i.e., TST device and Ag/Sb₂Te₃/Ag (ASA) device. As shown in supplementary figure S7, the conductivity of the TST device in its SET state increases with temperature, indicating the semiconducting property of the local filaments. In contrast, the ASA device shows negative correlation between its SET-state conductivity and temperature, indicating the formation of metallic Ag filaments as well known for such conventional types of devices”.

4) In addition to these experimental evidences, **COMSOL simulations** reveal that thermal confinement is stronger in Te filament than in conventional Ag filament, and the local temperature of Te filament under the condition of CC=1 mA reaches its melting temperature which is quite low.

5) In choosing the dielectric material, special consideration has been given to the composition of the material to avoid the possibilities of forming filaments other than Te. For instance, oxygen should be avoided due to the possibility of forming oxygen vacancies and therefore oxygen-poor filaments as widely observed in RRAM devices. Here, Sb₂Te₃ has been chosen due to its desired composition. The above experimental results from the Sb₂Te₃ device provide strong evidences that the reversible RS is enabled by Te filament formation and rupture.

Comment 5

The authors didn't address the question regarding the different off resistance values directly. It is not about the current compliance. Instead, the nonvolatile (Fig. 1c,e) mode generally shows lower OFF-state resistance than the volatile (Fig. 1d,f) mode. Will this contribute to the different switching behaviors as well?

Response

Thank you very much and we apologize for not responding to your comment on the issues related to the current compliance in our previous response letter. In the original manuscript, there is certain degree of difference in the presented OFF-state resistance between the NV-RS mode and V-RS mode. **We consider it as due to cycle-to-cycle variation rather than any indication of the intrinsic device property. As shown in our previously revised manuscript R1, such variation is clearly mitigated (figure 1).** This can probably be related to the slightly lower Sb_2Te_3 deposition rate used there. As supported by various experimental evidences obtained by device measurements and materials characterizations, the different switching behavior of the device under different compliance currents is believed to be due to a combination of unique electrical-thermal properties of Te filament. Statistical analysis of the device performance and its further optimization will be the subject to future works.

Reviewer #3

General comment

The manuscript has been improved along my comments and suggestions. The authors have also added new data to the revised manuscript, in response to the comments of other reviewers, which increases the quality of the paper. Thus, I recommend publication in Nature Communications. Please revise the following minor mistakes.

Response

Thank you very much again for spending your valuable time on reviewing our manuscript and providing professional suggestions to help further improve the quality of the manuscript.

Our responses to your specific comments one by one are shown as follows.

Comment 1

On page 4, line 122: '452' should be '452°C'.

Response

Thank you very much for pointing out this mistake. This problem has arose at the stage of .doc to .pdf conversion. In this revised manuscript R2, we have carefully checked it in its pdf format and avoided such problem.

Comment 2

The abbreviation of the Joule heating (JH) is denoted to 'Jh' in many places, such as line 277, 279, 285 of page 10, line 314 of page 11, line 363 of page 12, line 394 of page 13, line 401, 402 of page 14, etc.

Response

Thank you very much for pointing out this issue. In the previously revised manuscript R1, we have intended to use JH for Joule heating and Jh for Joule heat. As this is likely to cause confusion, in this revised manuscript R2 Joule heat is no longer abbreviated and Joule heating is abbreviated to JH.

Reviewers' Comments:

Reviewer #1:

Remarks to the Author:

The authors have made better responses and revisions in the newly submitted version of manuscript. I am overall satisfactory and can recommend it for publication.

Point-by-point response to reviewers' comments

Reviewer #1

General comment

The authors have made better responses and revisions in the newly submitted version of manuscript. I am overall satisfactory and can recommend it for publication.

Response

Thank you very much for recommending the publication of our manuscript, and again, for spending your valuable time on reviewing the manuscript.